# P-body proteins regulate transcriptional rewiring to promote DNA replication stress resistance

Raphael Loll-Krippleber[1] & Grant W. Brown [1]

mRNA-processing (P-) bodies are cytoplasmic granules that form in eukaryotic cells in response to numerous stresses to serve as sites of degradation and storage of mRNAs. Functional P-bodies are critical for the DNA replication stress response in yeast, yet the repertoire of P-body targets and the mechanisms by which P-bodies promote replication stress resistance are unknown. In this study we identify the complete complement of mRNA targets of P-bodies during replication stress induced by hydroxyurea treatment. The key P-body protein Lsm1 controls the abundance of *HHT1, ACF4, ARL3, TMA16, RRS1* and *YOX1* mRNAs to prevent their toxic accumulation during replication stress. Accumulation of *YOX1* mRNA causes aberrant downregulation of a network of genes critical for DNA replication stress resistance and leads to toxic acetaldehyde accumulation. Our data reveal the scope and the targets of regulation by P-body proteins during the DNA replication stress response.

[1] Department of Biochemistry and Donnelly Centre, University of Toronto, 160 College Street, Toronto, ON, Canada M5S 3E1. Correspondence and requests for materials should be addressed to G.W.B. (email: grant.brown@utoronto.ca)

DNA replication is influenced by both environmental and internal cues. Chemical agents or metabolic by-products can cause DNA modifications that stall or slow DNA replication forks[1]. Similarly, DNA secondary structures or the presence of the transcription machinery on the DNA can act as barriers to replication forks[1]. In order to fully and accurately replicate the genome in presence of these perturbants, eukaryotic cells encode a multi-faceted response referred to as the replication stress response[1]. Excess replication stress causes mutations, genome rearrangements, and loss of genetic material, and can result in cell death and disease[1, 2]. Therefore, DNA replication as well as the pathways controlling the response to stresses affecting this critical biological process must be tightly regulated.

In the model eukaryote *Saccharomyces cerevisiae*, the DNA replication stress response has been extensively studied at both the single-gene/protein level and at the genome/proteome scale[3–10]. DNA replication stress induces extensive changes in intracellular protein localization, with as much as 6% of the proteome re-localizing upon drug-induced DNA replication stress[8, 9, 11, 12]. Although much of the global protein location re-patterning that occurs during replication stress involves nuclear and intra-nuclear protein localizations, one of the most prominent type of localization change is the coalescence of proteins to form foci in the cytoplasm[8, 9]. Formation of cytoplasmic foci is a characteristic of mRNA-processing bodies (P-bodies) and indeed, the group of proteins that re-localize to cytoplasmic foci during replication stress is highly enriched in P-body components[8].

P-bodies are cytoplasmic membrane-free granules that contain non-translating mRNAs and proteins involved in mRNA decapping and degradation[13]. Critical P-body components include the decapping enzyme Dcp2, the decapping activator Dcp1, the decapping enhancers Edc1-3, Dhh1, Pat1 and Lsm1-7, and the 5′–3′ exonuclease Xrn1, which together determine the decapping or degradation rate of mRNAs[13]. P-bodies are present in non-stressed cells, however their number and size increase in response to diverse conditions including glucose starvation, osmotic stress, growth to stationary phase, and DNA replication stress[8, 14]. P-body formation during replication stress has unique genetic requirements[8], suggesting that P-body formation, and perhaps the regulation of mRNAs that are P-body targets, is a specifically controlled stress response. Furthermore, deletion of P-body genes confers sensitivity to DNA replication stress[8], and causes genome instability[15]. As available data indicate that P-body regulation of mRNA targets is relevant to DNA replication stress resistance, we sought to identify the network of P-body targets to understand how they contribute to cellular fitness during DNA replication stress.

Here, we define the network of mRNAs regulated by P-body proteins during DNA replication stress. We leverage the transcriptome data in combination with a systematic genetic suppression screen to identify the P-body targets that are important for DNA replication stress resistance. Our genomic analyses converge on the transcription factor Yox1, whose accumulation is toxic in cells experiencing replication stress, and whose mRNA localizes to P-bodies. Yox1 toxicity is likely due to its transcriptional repression function, as we found that Yox1 downregulates genes that are critical for resistance to replication stress. In particular, we show that *ALD6* de-repression is critical to prevent the toxic accumulation of acetaldehyde. Thus, we identify a key DNA replication stress resistance pathway regulated by the P-body target *YOX1*.

## Results

**Deletion of *LSM1* extensively remodels the transcriptome.** *LSM1* is essential for the regulation of functional P-body

formation, and for the degradation or stability of specific mRNA targets[16–18]. In order to identify the complement of mRNA targets of P-body regulation, we used RNA-seq to profile the transcriptomes of wild-type (WT) and *lsm1Δ* cells, in the presence and absence of the replication stress inducing drug hydroxyurea (HU) (Fig. 1a). The experiment was performed in duplicate on total RNA depleted of rRNA, and differential expression was assessed using the Tuxedo protocol[19]. Genes showing differential expression with a statistical *P*-value less than 0.05 after Benjamini–Hochberg correction[19] were considered as hits in our analysis. The transcriptome was greatly affected upon deletion of *LSM1* (Fig. 1b, c; Supplementary Data 1). As expected, given the role of Lsm1 in RNA degradation[16–18], we found that 333 mRNAs increased in abundance when *LSM1* was deleted. Unexpectedly, a similar number of mRNAs (258) decreased in abundance (Fig. 1b, c; Supplementary Data 1). Although we have not determined whether these mRNAs are direct P-body targets, it is possible that association of mRNAs with Lsm1 is in some cases protective and prevents exosome-dependent degradation[20]. Alternatively, absence of *LSM1* could stabilize transcriptional repressors, resulting indirectly in mRNA abundance decreases, as has been observed in cells lacking the 5′–3′ RNA exonuclease Xrn1[21].

In WT cells, HU treatment induced considerable change in mRNA abundance. Hundreds of transcripts showed increased or decreased abundance after only 1 h of HU treatment (Supplementary Fig. 1 and Supplementary Data 1). After 4 h of HU treatment 1134 and 1178 transcripts showed increased or decreased abundance, respectively (Supplementary Fig. 1). We found reasonable positive correlations between our data and previous microarray transcriptomic studies in HU[6] or methyl methanesulfonate (MMS)[7, 22] (Supplementary Fig. 2a, b). The correlation with data obtained using a distinct replication stress agent, MMS, indicates that a substantial fraction of the transcriptional program that we identified is due to DNA replication stress (Supplementary Fig. 2b). Some changes in gene expression that we observe are likely the result of cell-cycle synchronization in early S phase that occurs upon HU addition. Both WT and *lsm1Δ* undergo similar HU arrests, however, and so cell-cycle effects are unlikely to impinge on the identification of differentially expressed genes when WT and *lsm1Δ* are compared (Supplementary Fig. 3).

Deletion of *LSM1* had a dramatic effect on mRNA abundance during HU treatment. Between 499 (1 h after HU treatment) and 1203 transcripts (4 h after HU treatment) increased in abundance in the *lsm1Δ* strain (Fig. 1b, c and Supplementary Data 1) compared to the WT control. The unexpected transcript abundance decreases that we observed in the absence of HU treatment in *lsm1Δ* were more extensive in the presence of HU, where 322 to 1051 mRNAs decreased in abundance. Almost half of the differentially expressed genes in *lsm1Δ* cells were downregulated, again indicating an unanticipated positive or protective role for P-body proteins in the regulation of gene expression (Fig. 1b, c and Supplementary Data 1). Overall, however, the distribution of the mRNA abundance ratios for the differentially expressed genes showed a statistically supported positive bias, indicating that the stabilizing effect on RNAs observed upon deletion of *LSM1* is stronger than the unexpected destabilizing effect (Fig. 1d). Interestingly, a large fraction (as much as 53%) of the transcripts whose abundance was affected by HU treatment and deletion of *LSM1* were also affected by MMS treatment, suggesting that Lsm1-regulated transcripts change in abundance during DNA replication stress in general (Supplementary Fig. 2c).

Finally, to confirm that the differentially expressed genes that we identified were independent of the data analysis

method used, we applied two additional analyses to identify differentially expressed genes: EBSeq[23] and edgeR[24]. Between 34 and 79% of the genes identified in our initial analysis were also identified using EBSeq or edgeR, depending on the time point analyzed (Supplementary Table 1).

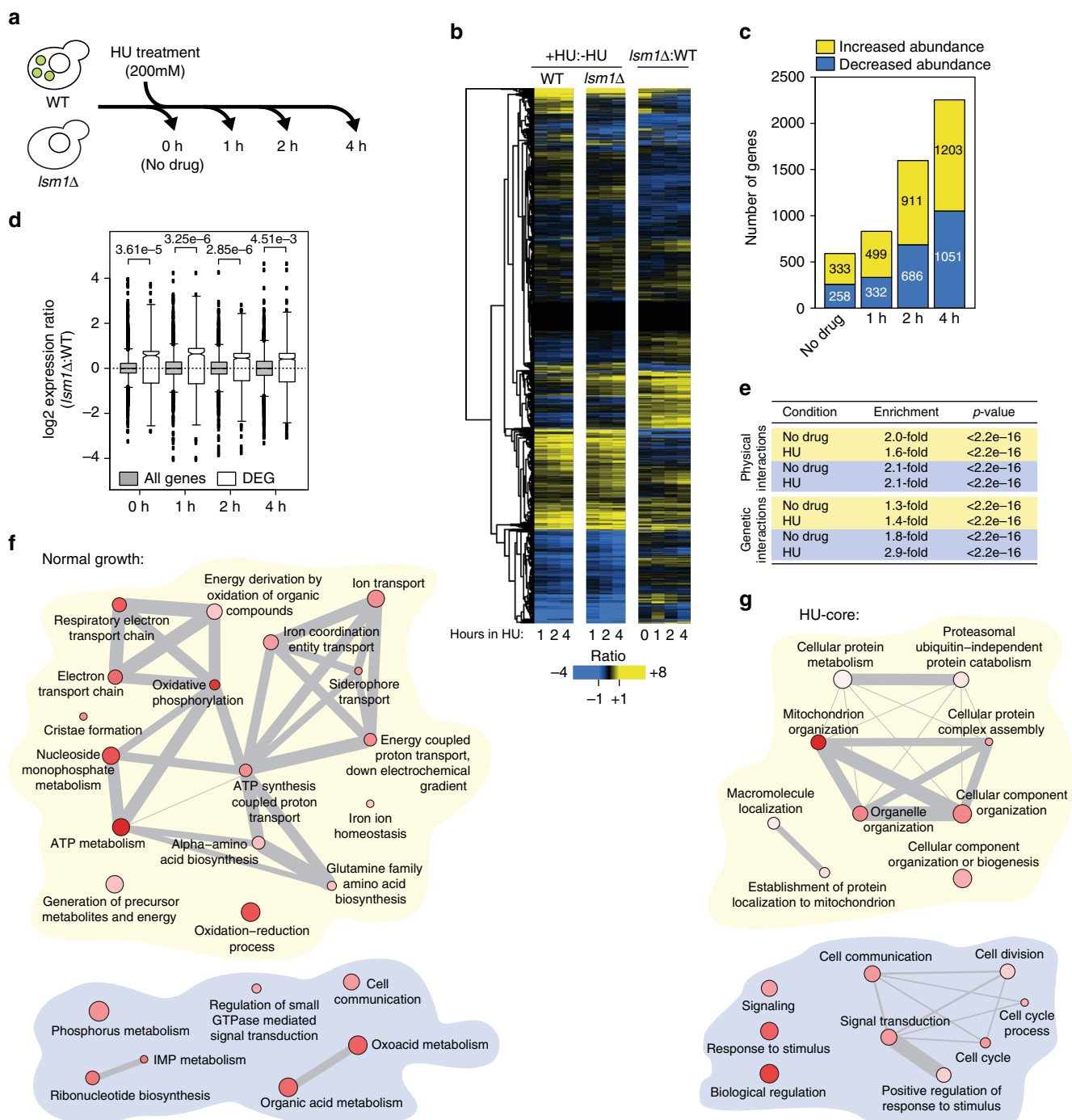

**Fig. 1** The transcriptome is extensively remodeled when functional P-bodies are absent. **a** Overview of the RNA-seq strategy used to measure RNA abundance in *lsm1*Δ cells. **b** Gene expression log2 ratios plotted as a heatmap. Expression ratios comparing HU treatment to the untreated condition (+HU: −HU), in wild type (*WT*) and *lsm1*Δ, are plotted on the *left*. Expression ratios comparing *lsm1*Δ cells to WT cells during normal growth and exposure to HU (*lsm1*Δ:WT) are plotted on the *right*. **c** The number of differentially expressed transcripts in *lsm1*Δ during replication stress is plotted. **d** The distributions of the *lsm1*Δ:WT expression ratios are displayed as boxplots for all genes (*gray*) and the differentially expressed genes (*white*). *P*-values are from unpaired Wilcoxon tests. **e** Summary of interaction enrichments for mRNAs with increased abundance in *lsm1*Δ (*yellow*) and with decreased abundance in *lsm1*Δ (*blue*) during normal growth (no drug) and during replication stress (HU). *P*-values were calculated using a binomial test. **f** GO term enrichment networks for mRNAs that were differentially expressed in *lsm1*Δ during normal growth, for mRNAs with increased abundance (*yellow*) and mRNAs with decreased abundance (*blue*). Nodes are shaded according to *P*-values (*darkest red* indicating the smallest *P*-value). Node size indicates the frequency at which a given GO term is found in the *S. cerevisiae* genome. Edges connect highly similar GO terms and *edge thickness* represents degree of similarity. **g** GO term enrichment networks for mRNAs that were differentially expressed in *lsm1*Δ during HU treatment (HU-core genes)

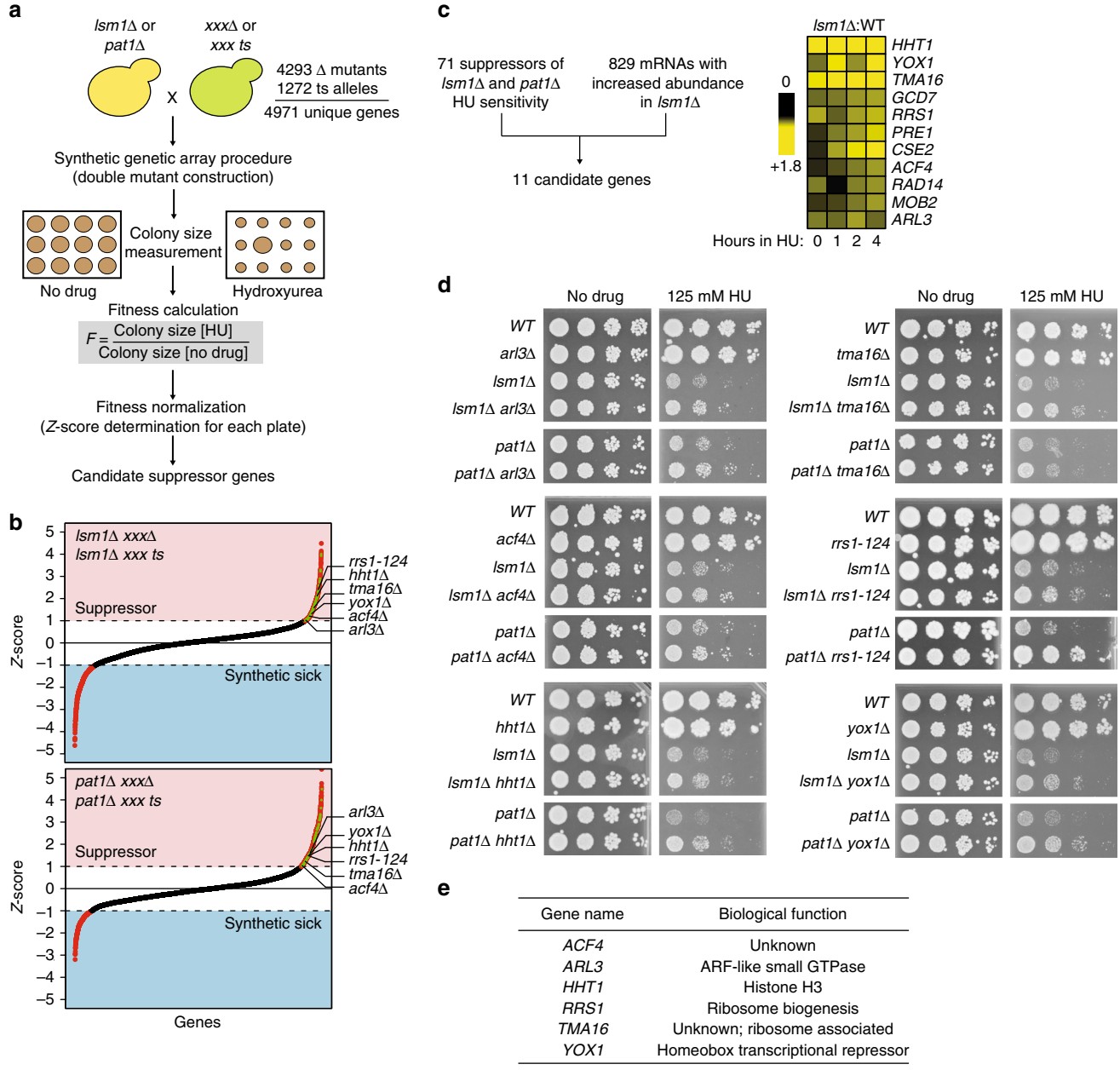

**Fig. 2** Combining a systematic suppressor screen with mRNA abundance data to identify P-body targets in replication stress. **a** Overview of the screen to identify suppressors of the HU sensitivity of P-body mutants. **b** Z-score normalized HU fitness values for the *lsm1Δ xxx* and *pat1Δ xxx* double mutants. Genes that were identified in both screens are indicated in *green*. Validated hits are indicated on the *right*. **c** The mRNA abundance during replication stress is displayed as a heatmap for eleven candidate genes from the intersection of the suppressor screen and mRNA abundance data. The log2 *lsm1Δ*:WT ratio is plotted for the indicated time points. **d** HU sensitivity assay for the 6 validated candidate genes among the 11 genes identified in **c**. Ten-fold dilutions of an overnight culture were spotted on YPD medium with or without HU. **e** Biological functions of the validated candidate genes

## Differentially expressed genes are functionally enriched.

Mining existing databases[25] revealed modest enrichments for both genetic and physical interactions in differentially expressed genes in *lsm1Δ*, both during normal growth and in DNA replication stress (Fig. 1e). These data indicate that genes that are differentially expressed in *lsm1Δ* relative to WT are more likely to share functional biological connections, and this is true both during unperturbed growth and in the presence of replication stress.

We further assessed functional connections among the differentially expressed genes by testing for enrichment of gene ontology (GO) biological process terms. Among the genes with increased mRNA abundance in *lsm1Δ* we noted enrichment for GO terms related to ATP synthesis, particularly genes encoding components of the electron transport chain, suggesting a role for P-bodies in regulating energy transactions during normal cellular growth (Fig. 1f and Supplementary Data 2). Beside ribonucleotide metabolism, little additional functional enrichment was observed among the genes with decreased expression in *lsm1Δ*, suggesting that mRNA abundance decreases in the absence of functional P-bodies during normal cell growth could be largely non-specific.

To assess functional enrichment in the presence of replication stress, we defined a core set of HU-responsive genes as the genes being differentially expressed only upon HU exposure and at two consecutive time points (601 increased abundance and 417 decreased abundance). In addition, we removed genes known to

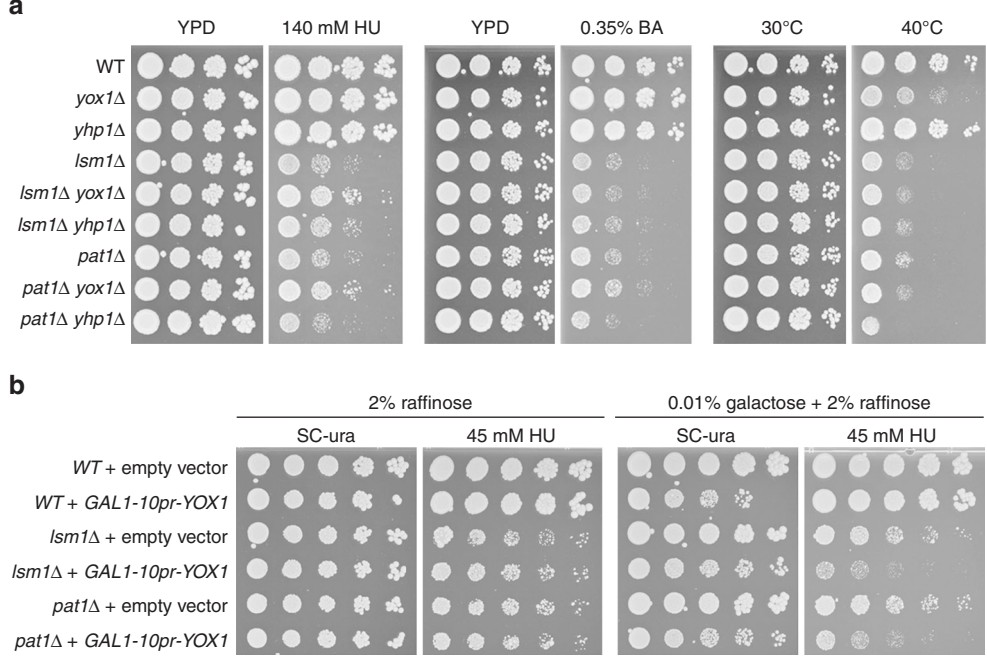

**Fig. 3** *YOX1* toxicity is specific to DNA replication stress. **a** Serial 10-fold dilutions of the indicated strains were spotted on YPD with or without HU, boric acid (BA) or at high temperature (40 °C). Plates were imaged after 2–3 days of growth to assess the sensitivity of each strain to each treatment. **b** Serial 5-fold dilutions of WT (BY4741), *lsm1Δ* or *pat1Δ* cells containing pBY011 (empty vector) or pBY011-*YOX1* (GAL1-10pr-*YOX1*) plasmids were spotted on synthetic complete medium lacking uracil and containing raffinose or raffinose and galactose, with or without HU. Plates were imaged after 7 days of growth to assess the effect of *YOX1* expression on HU sensitivity

be involved in the general environmental stress response from our core set of HU-responsive genes (72 genes in the increased abundance set (42 upregulated in the environmental stress response) and 49 genes in the decreased abundance set (35 downregulated in the environmental stress response))[26]. Genes engaged in several aspects of mitochondrial function, particularly protein import into the mitochondria, were enriched in the set of genes whose mRNA abundance increased in HU in *lsm1Δ*, as were genes encoding proteasome subunits (Fig. 1g and Supplementary Data 2). Genes encoding cell-cycle regulators were enriched in the genes whose mRNA abundance decreased in HU in *lsm1Δ* cells (Fig. 1g and Supplementary Data 2). The functional enrichments suggest that P-bodies downregulate mitochondrial biogenesis and protein degradation and upregulate genes involved in controlling cell-cycle progression during HU-induced replication stress.

Thus, in addition to the major regulatory effect that functional P-bodies exert on gene expression in unperturbed cells, we define an additional layer of regulation of transcript abundance by P-body proteins, both positive and negative, that is evident when replication stress is present. When compared to unperturbed cells, P-body proteins regulate different cellular processes during exposure to replication stress, consistent with a specific biological role for P-bodies in the DNA replication stress response.

**Suppressors of the HU sensitivity of P-body mutants.** Having established that the P-body protein Lsm1, either directly or indirectly, regulates the abundance of mRNAs during replication stress, we sought to identify mRNAs whose downregulation at P-bodies is important for replication stress resistance. We reasoned that the HU sensitivity of P-body mutants is due to the increased abundance of critical P-body target mRNAs. Therefore, deletion of the genes encoding these targets should suppress the

HU sensitivity of P-body mutants (Fig. 2a). Since *LSM1* and *PAT1* share the most genetic interactions of any two P-body components[27], we conducted suppression screens of both *lsm1Δ* and *pat1Δ* (Fig. 2b and Supplementary Data 3), using the Synthetic Genetic Array (SGA) method[28] to systematically inactivate 4971 genes (~76% of the genome) in an *lsm1Δ* or *pat1Δ* background. We assessed the fitness of every double mutant, by measuring and comparing colony size in presence and absence of HU, in triplicate. Fitness values were normalized in order to remove plate and/or pinning artefacts, and converted to Z-scores. The Z-score distribution was similar in both *lsm1Δ* and *pat1Δ* strain backgrounds (Fig. 2b). Double mutants with a Z-score greater than +1 were considered to be HU sensitivity suppressors. Because the individual suppressor screens had a low true positive rate (~27%; 3 candidates validated, after strain re-construction, out of 11 randomly picked suppressor hits) we limited our analysis to suppressors identified in both the *lsm1Δ* and *pat1Δ* screens. A total of 71 genes, when inactivated, were able to suppress both *lsm1Δ* and *pat1Δ* HU sensitivity. Among these candidate P-body target genes, 11 also have increased transcript abundance in the absence of *LSM1* at 2 or more consecutive times after HU treatment (Fig. 2c). Independent re-construction of the *pat1Δ* and *lsm1Δ* double mutants with each of the 11 genes resulted in validation of 6 putative target genes: *ARL3*, *ACF4*, *HHT1*, *TMA1*, *RRS1* and *YOX1* and increased mRNA abundance in HU for 5 of these transcripts was confirmed by two independent data analysis methods. *ACF4* was confirmed by edgeR but not by EBSeq (Supplementary Table 2). Mutations in each suppressed both *lsm1Δ* and *pat1Δ* HU sensitivity (Fig. 2d). In no case was suppression complete, suggesting that multiple P-body targets contribute to fitness during the replication stress response. The biological function of the 6 genes is described in Fig. 2e. Two genes (*RRS1* and *TMA16*) encode ribosome-associated proteins[29, 30], suggesting that downregulation of protein translation capacity is important for DNA replication stress resistance.

Two (*ACF4* and *YOX1*) encode phosphorylation targets of the cyclin dependant kinase Cdc28[31]. Importantly, among the six genes we identified, one (*HHT1*) is known to be a P-body target during replication stress[32]. Therefore, our strategy identified five new genes whose expression is toxic during replication stress when P-body function is compromised.

**YOX1 regulation by P-bodies is critical during HU stress.** Yox1 is a homeobox transcriptional repressor that regulates the expression of cell-cycle regulated and DNA replication genes[33]. Together with Mcm1, Yox1 binds to the promoters of genes whose regulation is critical during the M/G1 transition of the cell cycle[33]. Inactivation of the Yox1 protein is important

during replication stress in *Schizosaccharomyces pombe*[34], and overexpression of *YOX1* causes a dramatic increase in chromosome loss in *S. cerevisiae*[35]. DNA replication stress causes Yox1 to re-localize from the nucleus to the cytoplasm[8], suggesting that expression of Yox1 targets could play an important role in the replication stress response. Given its connections to the replication stress response, we asked if *YOX1* expression is regulated by P-body proteins.

We first confirmed that *YOX1* mRNA levels increase in P-body mutants. When we examined *YOX1* mRNA by quantitative reverse transcription PCR (RT-PCR), we found that in both *lsm1*Δ and *pat1*Δ *YOX1* mRNA accumulated relative to the WT cells (Supplementary Fig. 4). Deletion of *PAT1* abolishes P-body formation during replication stress[8], while loss of *LSM1*

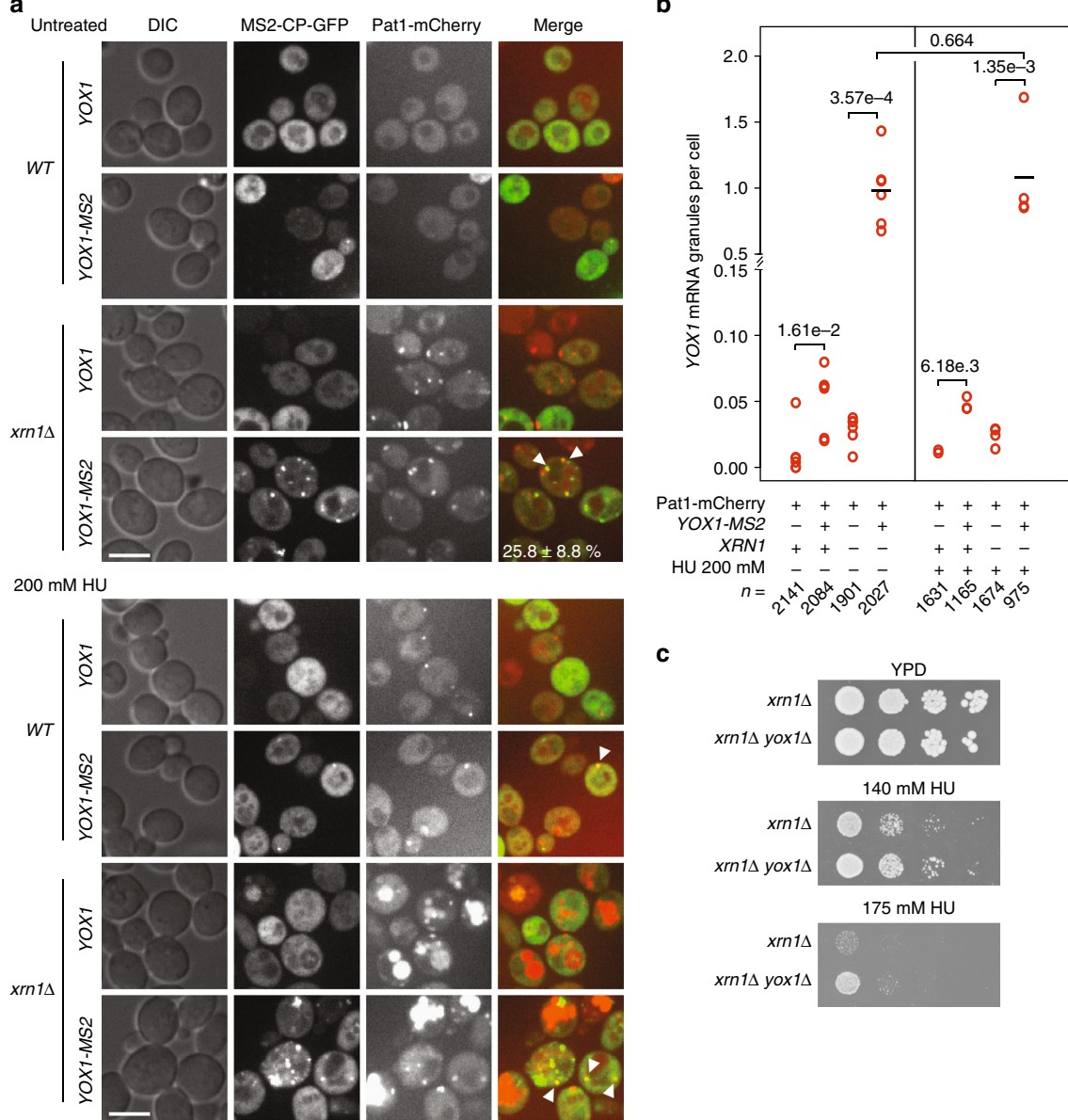

**Fig. 4** *YOX1* mRNA forms cytoplasmic granules and accumulates at P-bodies. **a** Representative images of cells co-expressing MS2-CP-GFP and Pat1-mCherry, with or without *YOX1* tagged with MS2 repeats, before and after 2 h of treatment with HU. Cytoplasmic granules were assessed in both WT and in *xrn1*Δ cells. *Scale bars* are 7 μm. *Arrows* indicate *YOX1* mRNA granules that co-localize with Pat1-mCherry granules. **b** Quantification of *YOX1* mRNA granules in untreated and HU-treated cells. The average number of foci per cell is plotted for each of at least three replicates. The solid bar indicates the average of the replicates, *P*-values are from Student's *t*-tests, and *n* indicates the total number of cells examined. **c** Serial 10-fold dilutions of the indicated strains were spotted on YPD with or without HU. Plates were imaged after 3–4 days of growth to test the suppression of *yox1*Δ HU sensitivity by deletion of *XRN1*

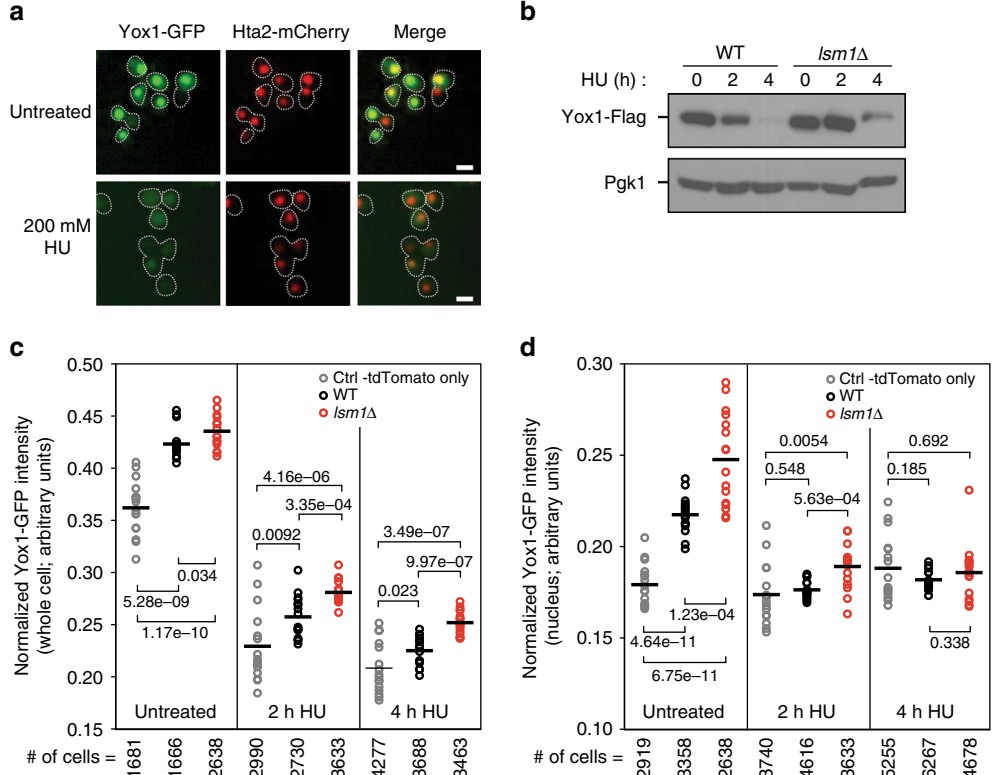

**Fig. 5** The Yox1 repressor is more abundant in cells lacking LSM1. **a** Representative images of cells expressing Hta2-mCherry and Yox1-GFP before and after 2 h of HU treatment. The *scale bars* are 5 μm, and the outlines of the cells are indicated. **b** Analysis of Yox1-Flag protein level by immunoblot before and after HU treatment. Pgk1 was used as loading control. Molecular weight markers are shown on the uncropped image in Supplementary Fig. 5. One representative experiment is shown. Normalized intensity of the cellular (**c**) or the nuclear (**d**) Yox1-GFP fluorescence in WT and *lsm1*Δ cells. Each point represents the data for an individual replicate of the indicated strain (for 8 technical replicates for each of 2 biological replicates). The *solid bar* indicates the average of the replicates, and *P*-values from the Student's *t*-test are indicated

results in the accumulation of non-functional P-bodies[18], suggesting that *YOX1* degradation is, to some extent, dependent on the formation of functional P-bodies during replication stress. However, because Lsm1-dependent and Pat1-dependent mRNA degradation can also occur in absence of visible P-body granules, it is possible that *YOX1* mRNA regulation by P-body proteins could occur at sites external to visible P-body granules.

We next tested whether *YOX1* inactivation rescued the sensitivity of P-body mutants to diverse stresses or was specific to DNA replication stress induced by HU treatment (Fig. 3). Deletion of *LSM1*[36] or *PAT1* confers sensitivity to boric acid, yet deletion of *YOX1* failed to rescue either sensitivity (Fig. 3a). The *lsm1*Δ mutant is also sensitive to heat stress[37], and temperature sensitivity was not rescued by *yox1*Δ (Fig. 3a). The *YOX1* gene has a paralog, *YHP1*, so we also assessed the effect of deleting *YHP1* on stress resistance in *lsm1*Δ and *pat1*Δ (Fig. 3a). Deletion of *YHP1* had little effect on the HU, heat, or boric acid sensitivity of the P-body mutants. In addition, *YHP1* mRNA was not upregulated in *lsm1*Δ (Supplementary Data 1). We conclude that deletion of *YOX1* rescues the HU sensitivity of P-body mutants while playing little role in other P-body stress responses, and that the paralog *YHP1* is unlikely to be a relevant P-body target during replication stress.

We reasoned that if P-body-dependent degradation of *YOX1* mRNA is critical for replication stress resistance, then over-expression of *YOX1* should sensitize P-body mutant strains to HU. Indeed, expression of *YOX1* from the *GAL1-10* promoter was toxic in both *lsm1*Δ and *pat1*Δ backgrounds (Fig. 3b). Together these data support the conclusion that negative

regulation of *YOX1* expression by P-body proteins is important specifically for HU-stress resistance.

**The *YOX1* transcript localizes to P-bodies.** To assess whether the negative regulation of *YOX1* expression was likely occurring directly at the cytoplasmic P-body granules that form during replication stress, we tracked the location of *YOX1* mRNAs in live cells using the mTAG mRNA visualization system[38]. We introduced MS2 repeats into the 3′ UTR of the *YOX1* gene and expressed the MS2 coat protein fused to green fluorescent protein (GFP) to visualize the *YOX1* mRNA. P-bodies were visualized simultaneously by co-expressing a Pat1-mCherry fusion protein. The MS2 tag system can compromise the visualization of full-length mRNAs by promoting Xrn1-dependent accumulation of mRNA degradation products[39, 40], so we examined *YOX1* mRNA localization in both WT and in cells deleted for *XRN1*. Additionally, we reasoned that because the *YOX1* transcript is potentially degraded at P-bodies where Xrn1 is the predominant 5′–3′ exoribonuclease[41], deletion of *XRN1* should facilitate *YOX1* mRNA visualization by stabilizing it.

*YOX1* mRNA forms granules in ~6% of WT cells, both in untreated conditions and after HU treatment (Fig. 4a, b). Deletion of *XRN1* dramatically increased the formation of *YOX1* mRNA granules and of P-bodies[18] in untreated conditions and after HU treatment (Fig. 4a, b). Control strains expressing the MS2-GFP coat protein but lacking the MS2 repeats in the 3′UTR of *YOX1* displayed very few granules (Fig. 4b), demonstrating that MS2-GFP focus formation required *YOX1* tagged with MS2

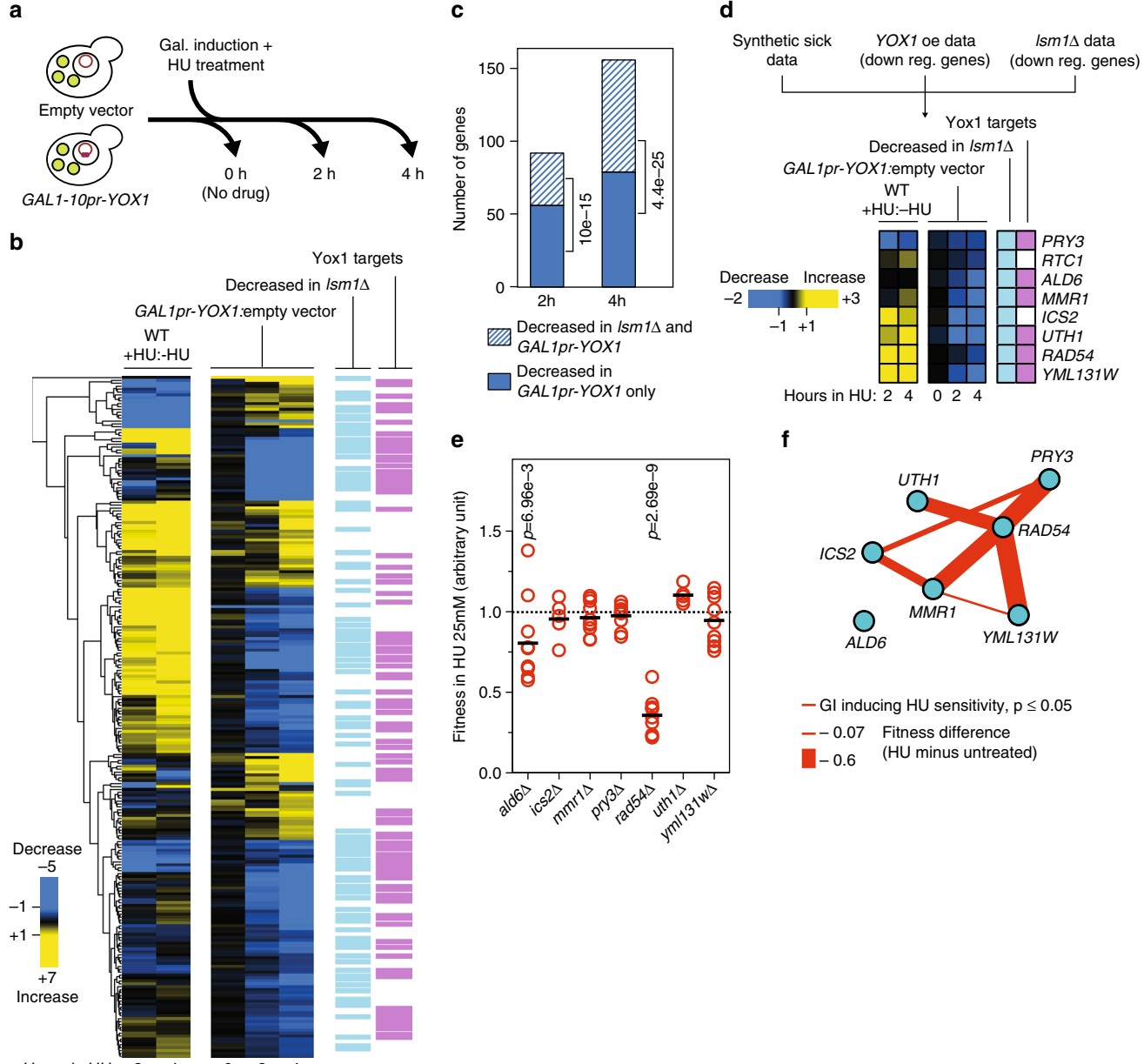

**Fig. 6** Excess Yox1 represses genes that are critical for cell fitness during replication stress. **a** Overview of the RNA-seq experiment. WT cells containing pBY011 (empty vector) or pBY011-*YOX1* (*GAL1-10pr-YOX1*) vectors were cultivated in synthetic media containing raffinose. Overexpression of *YOX1* was induced by adding galactose and replication stress was induced by HU treatment. RNA was extracted at the indicated time points. **b** Heatmap of expression ratios generated by hierarchical clustering, showing genes that were differentially expressed upon *YOX1* overexpression. Two different ratios are represented: +HU:−HU compares expression values in the control strain (empty vector) between the HU condition and the untreated condition, and *GAL1-10pr-YOX1*:empty vector compares expression values between the *YOX1*-overexpressing strain and the control strain at each time point. *Light blue bars* indicate genes that were downregulated after 2 h and/or 4 h in HU in *lsm1Δ* compared to WT (as in Fig. 1). *Purple bars* denote genes that are described as Yox1 targets in the Yeastract database. **c** The number of genes with decreased mRNA abundance upon *YOX1* overexpression, with the overlap of genes that also have decreased mRNA abundance in *lsm1Δ* is plotted as a stacked bar graph. The indicated *P*-values were calculated using a hypergeometric test. **d** Identification of potential Yox1 targets during replication stress at the intersection of three datasets. Heatmaps are as presented in **b**. **e** Fitness data in HU for the single mutants identified in **d**. Each data point represents a single replicate (*n* = 5). Horizontal bars represent the average of the replicates. *P*-values are from a Student's *t*-test. **f** Condition-dependent negative genetic network of interactions between the 8 targets in **d**. *Red edge thickness* is proportional to the difference between the double mutant fitness in absence and presence of 25 mM HU

repeats. Importantly, the *YOX1* granules co-localized with the P-body marker Pat1 in both WT and *xrn1Δ* cells under normal conditions and after HU treatment (Fig. 4a, b). In 1092 *YOX1* granules examined, 25.8 ± 8.8% of *YOX1* mRNA and Pat1 granules co-localized in *xrn1Δ* untreated cells. Similar levels of co-localization were found with other transcripts and the P-body

marker Edc3[42]. HU treatment resulted in the appearance of large aggregates in the red channel, making the quantification of *YOX1* mRNA granule co-localization with Pat1-mCherry difficult to evaluate (Fig. 4a). Yet, when discreet Pat1 foci were observable we could detect co-localization with *YOX1* mRNA granules (Fig. 4a). Interestingly, the accumulation of *YOX1* mRNA granules that is

evident upon deletion of *XRN1* correlates with increased HU sensitivity (Fig. 4c) and increased *YOX1* transcript abundance (*xrn1*Δ:wildtype = 3.7 ± 1.8). As was the case with *lsm1*Δ, the HU sensitivity of *xrn1*Δ is rescued by deletion of *YOX1* (Fig. 4c), indicating that *YOX1* mRNA is a critical transcript for HU

sensitivity of *xrn1*Δ. Thus, failure to degrade *YOX1* mRNA, either at the level of functional P-body formation or at the level of nucleolytic degradation of the mRNA, results in HU sensitivity. These data reinforce the notion that control of

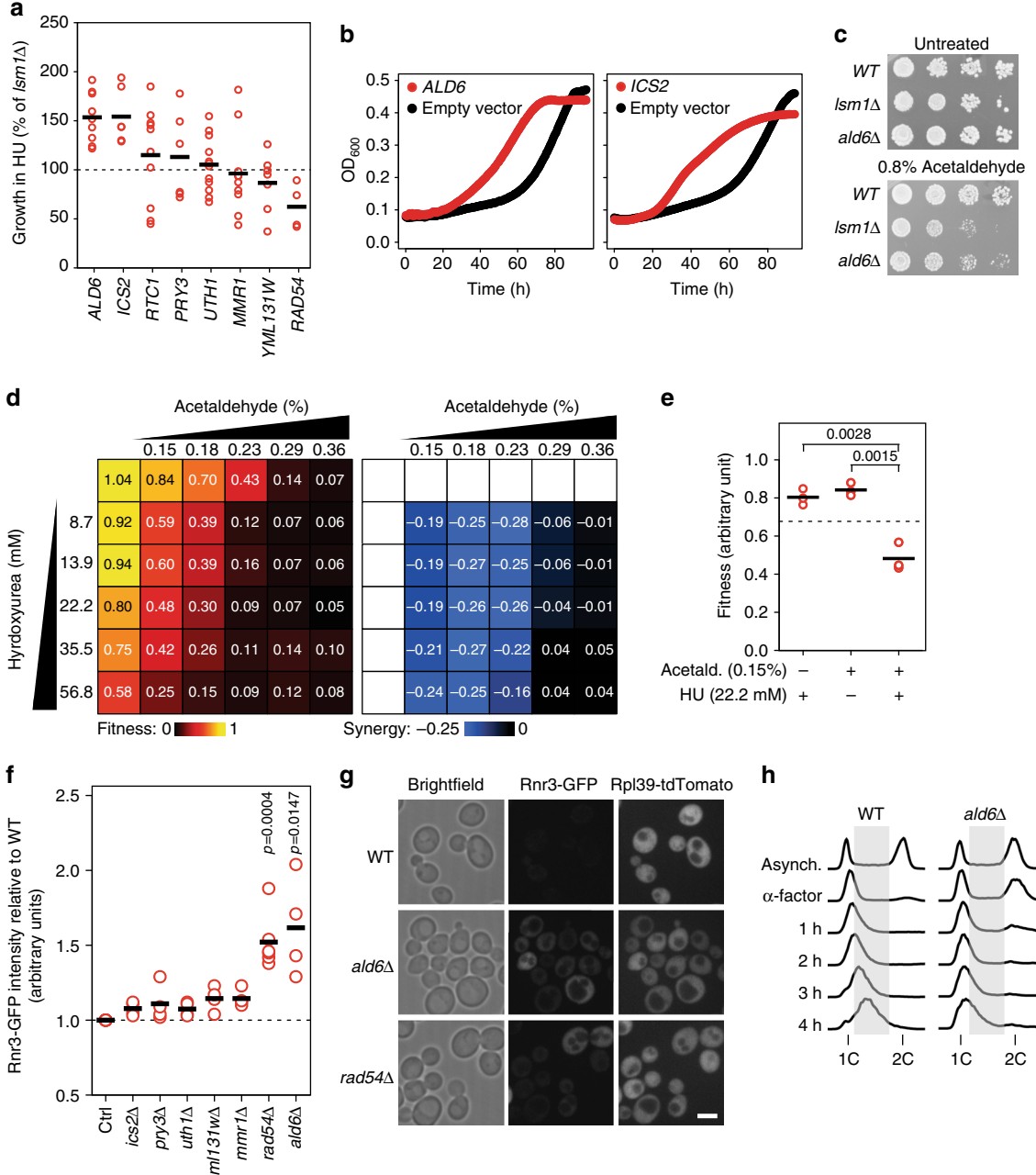

**Fig. 7** De-repression of the Yox1 target *ALD6* is critical to detoxify acetaldehyde and promote DNA replication stress resistance. **a** Relative growth of *lsm1*Δ cells overexpressing the indicated genes compared to *lsm1*Δ cells containing the control vector. Each data point represents a single replicate. Each experiment was repeated at least three times and included two independent biological replicates each time. The *solid bar* indicates the average. **b** Growth curves of *lsm1*Δ cells overexpressing *ALD6* or *ICS2* (*red*) or *lsm1*Δ cells containing the control vector (*black*) in the presence of HU. **c** Serial 10-fold dilutions of the indicated strains were spotted on YPD with or without acetaldehyde. Plates were imaged after 3–4 days of growth. **d** *Left*: Heatmap of fitness values of cells grown in presence of acetaldehyde, hydroxyurea or combinations of both drugs. Fitness values are relative to the untreated condition, for three independent replicates. *Right*: Synergy heatmap. Indicated values correspond to the difference between the observed fitness in the given drug combination and the expected fitness. **e** Fitness of cells grown in presence of acetaldehyde, hydroxyurea or combination of both drugs at the indicated concentrations. Each data point represents one replicate. *P*-values are from a Student's *t*-test. **f** Rnr3-GFP intensities in the indicated mutants relative to the WT. Each data point represents one replicate. *P*-values are from a Student's *t*-test. **g** Representative microscopic images of the indicated strains expressing Rnr3-GFP and RPL39pr-TdTomato. *Scale bar*: 5 µm. **h** Cell-cycle progression of the WT and *ald6*Δ strain after alpha-factor block and release in YPD containing 200 mM of HU

*YOX1* mRNA abundance is important to the replication stress response.

Together, these data indicate that *YOX1* mRNAs localize to P-bodies both in normal growth and during replication stress induced by HU, consistent with a model in which *YOX1* mRNA abundance is regulated by P-body proteins, including Xrn1, directly at P-bodies.

**Yox1 accumulates in the nucleus in *lsm1Δ* cells.** Yox1 and its paralog Yhp1 bind to the transcriptional antirepressor Mcm1, resulting in repression of Mcm1 targets[33]. Having established that the *YOX1* transcript increases in *lsm1Δ* cells in the presence of HU (Fig. 2c), we tested whether the Yox1 protein is increased in abundance as well. Yox1 accumulates in the cytoplasm upon HU treatment[8], so we examined Yox1-GFP protein abundance in individual cells and in individual nuclei in both WT and *lsm1Δ* cells (Fig. 5). Cells lacking *LSM1* had elevated levels of Yox1-GFP, in the nucleus and in the cytoplasm (Fig. 5c, d), consistent with the increased *YOX1* mRNA abundance in *lsm1Δ*. The increased level of Yox1-GFP in the nucleus in *lsm1Δ* persisted upon HU treatment for at least 2 h (Fig. 5d). Immunoblot analysis of Flag-tagged Yox1 confirmed an increase of Yox1 protein in *lsm1Δ* cells (Fig. 5b). It is worth noting that Yox1 protein abundance did not precisely mirror *YOX1* mRNA abundance, suggesting additional regulation at the level of translation and/or protein stability. These data indicate that there is more Yox1 in the nucleus when the *YOX1* mRNA is not degraded effectively at P-bodies, and suggest that Yox1 target genes will be expressed at lower levels when P-body function is compromised.

**The network of Yox1 targets is important for HU resistance.** To probe the consequences of increased *YOX1* expression during replication stress, we used RNA-seq to measure transcript abundance during *YOX1* overexpression in the presence of HU (Fig. 6a, b and Supplementary Data 4). The expression of between 96 (at 2 h) and 156 genes (at 4 h) was decreased in the *YOX1*-overexpressing strain. One third (2 h) to one half (4 h) of the genes that were downregulated upon *YOX1* overexpression were also downregulated in the *lsm1Δ* cells, consistent with modulation of Yox1 function by Lsm1 (Fig. 6b, c). This effect is statistically supported as *lsm1Δ* downregulated genes were strongly enriched among the downregulated genes identified upon *YOX1* overexpression (Fig. 6c). Interestingly, a number of the genes with altered expression in the presence of HU and *YOX1* over-expression were not known Yox1 targets[43] (Fig. 6b, *purple panel*). Therefore the potential repertoire of Yox1 regulatory targets is expanded during replication stress.

Surprisingly, the *YOX1* transcriptional repression program that we identified was not enriched for genes involved individually in HU resistance (hypergeometric $P = 0.992$ at 2 h and $P = 0.998$ at 4 h). We focussed on the 8 Yox1 targets that were also downregulated in *lsm1Δ* and that caused HU synthetic sickness in *lsm1Δ* (Fig. 6d), reasoning that the most critical Yox1 targets would be affected by Yox1 overexpression, *LSM1* deletion, and would exacerbate the HU sensitivity of *lsm1Δ*. We noted that only two (*ALD6* and *RAD54*) were sensitive to DNA replication stress on their own (Fig. 6e). To test the possibility that redundancy could account for the relative absence of single-mutant HU sensitivity, we constructed 20 combinations of double knockout mutants of the 8 targets (excluding *RTC1* which was not available in our deletion collection). We analyzed fitness by monitoring growth in liquid medium in presence or absence of HU. Using this approach, we identified 7 negative genetic interactions that were worsened by the presence of DNA replication stress (Fig. 6f, Supplementary Data 5). Three

interactions involved genes that were not HU sensitive as single mutants, and four involved *RAD54*. Therefore, functional redundancy among genes in the *YOX1* regulome can mask the roles of individual genes in replication stress resistance. We speculate that detailed analysis of double mutant space in the *YOX1* regulome would reveal many additional interactions that confer HU sensitivity.

Together, these data suggest that accumulation of Yox1 that occurs in the absence of a functional P-body pathway results in the downregulation of a transcriptional network which as a whole maintains cell fitness during replication stress.

**Acetaldehyde detoxification is critical upon HU exposure.** Although the *YOX1* transcriptional program as a whole is likely to contribute to replication stress resistance, we also tested whether we could detect roles for individual target genes. The 8 identified targets are downregulated upon *YOX1* overexpression and when P-body function is compromised (Fig. 6d). Therefore, we expressed each gene from the *GAL1-10* promoter in an *lsm1Δ* strain in the presence of HU to test whether any could individually rescue the HU sensitivity of *lsm1Δ* (Fig. 7a, b). Expression of *ICS2* or *ALD6* increased the fitness of *lsm1Δ* during replication stress (Fig. 7a, b) indicating that de-repression of *ICS2* and *ALD6* via P-body protein dependent degradation of *YOX1* mRNA contributes to replication stress resistance. We identified binding sites for both Yox1 and its co-repressor Mcm1 in the 1000-bp promoter regions of *ALD6* and *ICS2* using YeTFaSCo[44] (Supplementary Table 3), although it is also possible that both are indirect targets. Finally, *lsm1Δ* is sensitive to acetaldehyde (Fig. 7c), as would be expected if expression of the aldehyde dehydrogenase encoded by *ALD6* is reduced when P-bodies are not functional.

We probed the mechanism by which *ALD6* contributes to replication stress resistance. First we used quantitative RT-PCR to confirm that *ALD6* mRNA abundance decreases in *lsm1Δ* (Supplementary Fig. 6). Ald6 converts acetaldehyde into acetate, and thus *ald6Δ* cells accumulate higher amounts of acetaldehyde[45]. Acetaldehyde induces DNA inter-strand crosslinks that block DNA replication and transcription[46] and therefore could potentially be toxic in cells experiencing DNA replication stress. We found that HU and acetaldehyde were synergistic over a range of concentrations, indicating that the presence of acetaldehyde during HU-induced replication stress is indeed toxic (Fig. 7d, e). Deletion of *ALD6* causes strong constitutive S-phase checkpoint activation, as revealed by the induction of *RNR3* expression in *ald6Δ* cells to similar level as in *rad54Δ*, one of the strongest inducers of checkpoint-dependent *RNR3* expression[15] (Fig. 7f, g). Consistent with the accumulation of replication blocking DNA lesions in cells lacking Ald6, and the synergy between acetaldehyde and HU, we found that *ald6Δ* cells displayed slow S-phase progression in HU relative to WT cells.

Together, these data suggest that acetaldehyde accumulation causes endogenous replication stress that sensitizes cells experiencing additional sources of DNA replication stress. Therefore, de-repression of *ALD6* during HU-induced replication stress promotes fitness by removing acetaldehyde, reducing DNA damage, and reducing barriers to DNA synthesis.

**Discussion**

P-bodies form in the cytoplasm in response to a wide range of stresses, including DNA replication stress induced by HU[8, 14]. Several lines of evidence indicate that P-body function is important for the cellular replication stress response: At least two genes regulate P-body formation in replication stress but not in other stresses[8]. Deletion of P-body genes confers sensitivity to

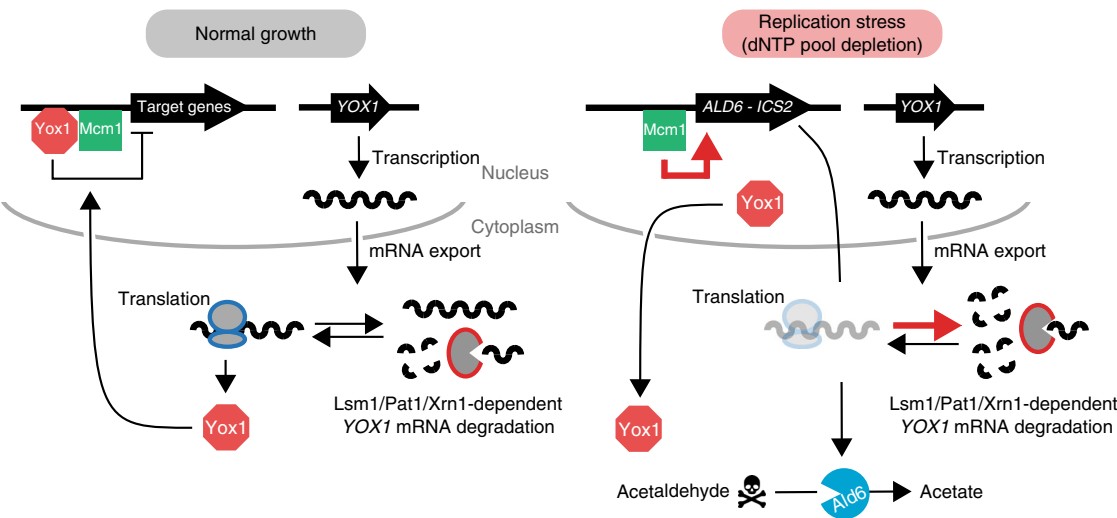

**Fig. 8** A model for *YOX1* post-transcriptional regulation by P-bodies during replication stress. During normal growth *YOX1* mRNA is produced and exported to the cytoplasm where it can either be degraded or translated. *YOX1* mRNA translation produces a transcriptional repressor that is imported to the nucleus where it binds the Mcm1 antirepressor to represses the transcription of target genes. During replication stress induced by hydroxyurea treatment, P-body formation is increased which results in increased *YOX1* mRNA degradation (*red arrow*). Export of the Yox1 protein from the nucleus concomitant with degradation of the *YOX1* mRNA allows the cells to control the amount of the Yox1 repressor in the nucleus. Decreased nuclear Yox1 relieves transcriptional repression (*red arrow*) of target genes required for HU resistance, including *ALD6* and *ICS2*. De-repression of *ALD6* promotes degradation of acetaldehyde, a molecule that is strongly toxic during replication stress

replication stress[8, 32]. One biologically relevant P-body target in HU, the gene encoding histone H3, has been identified[32]. Finally, P-body mutants display spontaneous DNA replication stress[15, 32]. Here we defined the P-body-regulated transcriptome, both during normal growth and in the presence of replication stress. We leveraged the RNA abundance data to identify five mRNA transcripts whose P-body regulation is important for replication stress resistance, *ARL3*, *ACF4*, *RRS1*, *TMA16* and *YOX1*. We demonstrated that *YOX1* upregulation is toxic in cells lacking a functional P-body pathway, and that *YOX1* mRNA localizes to P-bodies in live cells, consistent with direct degradation of *YOX1* mRNA at P-bodies. Finally, we showed that in the absence of proper regulation of *YOX1* mRNA by P-body proteins, Yox1 protein accumulates in the nucleus resulting in aberrant regulation of gene expression during replication stress. We propose a model (Fig. 8) where P-body formation is induced in response to replication stress, providing the cell with increased mRNA degradation capacity, resulting in a reduced pool of *YOX1* mRNA available for translation. Increased *YOX1* mRNA degradation, concomitant with the export of the Yox1 protein from the nucleus precludes the nuclear accumulation of the Yox1 repressor during replication stress. This orchestrated response allows cells to de-repress a program of gene expression, and individual genes, needed for replication stress survival.

Since mRNA degradation can occur in the absence of visible P-bodies[47, 48], it is possible that assembly of P-body proteins into microscopically visible structures is not required to regulate transcript abundance during replication stress. However, we find that at least three P-body proteins (Lsm1, Pat1, and Xrn1) regulate *YOX1* mRNA abundance. Loss of Lsm1 triggers accumulation of non-functional P-bodies in unstressed cells, as does loss of Xrn1, and prevents Pat1 recruitment to P-bodies during glucose starvation stress[18]. Loss of Pat1 also prevents functional P-body formation during DNA replication stress induced by HU[8]. In addition, we observed accumulation of *YOX1* mRNA cytoplasmic granules upon Xrn1 deletion, and many of the *YOX1* mRNA granules contained Pat1. Although we cannot exclude the possibility that regulation of *YOX1* mRNA by P-body proteins

occurs in the soluble fraction of the cytoplasm, our data are consistent with a model in which at least a portion of *YOX1* mRNA is degraded at P-bodies.

As expected, we identified a group of transcripts whose abundance increases in *lsm1Δ* cells, consistent with the possibility that some are degraded at P-bodies in WT cells. Surprisingly, we found an almost equal number of mRNAs whose abundance decreases in *lsm1Δ* cells. Interestingly, this paradoxical effect has been described in studies exploring the interconnection between transcription and mRNA degradation. For instance, the cytoplasmic mRNA exonuclease Xrn1 buffers transcription by activating the Nrg1 transcriptional repressor[21]. Although Nrg1 was not upregulated in *lsm1Δ* cells, the upregulation of the Yox1 repressor in *lsm1Δ* that we observe explains a fraction of the downregulated genes (~5%). Upregulation of other transcriptional repressors could further increase the number of genes downregulated in *lsm1Δ*, although we did not identify additional repressors in our dataset. Another possibility is that transcriptional activators are downregulated in *lsm1Δ* cells and therefore fail to activate their targets. We did find cases of downregulated transcriptional activators (Fkh2, Bas1, Rgt1, Pho2 and Leu3) in cells lacking Lsm1, and also found that the targets of these activators were enriched in the set of *lsm1Δ* downregulated genes (average hypergeometric $P = 2.7 \times 10^{-3}$ at 4 h in HU). Together, this phenomenon could account for an additional ~5% of the downregulated genes in *lsm1Δ* cells. Interestingly, mRNA degradation factors such as Xrn1, Dcp2, and Lsm1 can shuttle between the cytoplasm and the nucleus where they bind to gene promoter regions and stimulate transcription initiation and elongation[49], raising the further possibility that some fraction of the downregulated *lsm1Δ* genes are direct nuclear targets of Lsm1. Finally, given the magnitude of the transcriptional downregulation in the absence of *LSM1* it is possible that P-bodies can in some instances protect and stabilize mRNAs from exosome-dependent degradation[20], perhaps in concert with translational repression[50].

We find that *LSM1* is critical for the post-transcriptional regulation of mRNA abundance during replication stress. Exposure

to HU-induced replication stress causes massive mRNA abundance perturbations (this study and Dubacq et al.[6]), between 25% (1 h HU) and 45% (4 h HU) of which involve Lsm1 (and likely P-body) function. Of the mRNAs whose abundance was altered in a P-body-dependent manner during HU treatment (HU-responsive genes), only 11.8% were general environmental stress response genes, indicating that *LSM1* regulates much of the specific mRNA abundance response to replication stress. In total, combining deletion of *LSM1* with replication stress resulted in the accumulation of between 499 and 1203 transcripts. Distinct biological processes were enriched in the set of transcripts that accumulated in *lsm1Δ* during replication stress, suggesting that the response was specific and biologically significant. Although the primary target of HU is ribonucleotide reductase, inactivation of which limits dNTP pools to cause replication stress, there is evidence that HU affects other processes including iron metabolism[51, 52]. We have no doubt that some fraction of the *lsm1Δ*-affected HU transcriptome could reflect cellular events in addition to replication stress. However, we found that between 25 and 53% of affected genes (including *YOX1*) overlap with genes that are differentially expressed in a distinct replication stress agent, MMS, and that at least two of the effectors of the Yox1 transcriptional program, *RAD54* and *ALD6*, are individually important to prevent DNA replication stress. We conclude that the key elements of the *lsm1Δ*-affected HU transcriptome that we have deciphered are relevant to DNA replication stress. The functional targets of Lsm1 and P-bodies in other forms of stress, both chemical and genetic, remain to be determined.

In addition to the P-body-dependent regulation of *YOX1* and *HHT1* mRNAs, we identified 4 other P-body targets during replication stress: *ACF4*, *ARL3*, *TMA16* and *RRS1*. The P-body-dependent regulation of these genes is likely to contribute significantly to HU resistance, as their inactivation results in suppression of the HU sensitivity of P-body mutants comparable to that seen upon deletion of *YOX1* or *HHT1*. Of particular interest, two of the putative P-body targets have ribosomal functions. *RRS1* encodes an essential protein that binds to the 60S ribosomal subunit and is also required during ribosome biosynthesis[29]. *TMA16* encodes a protein of unknown function that associates with ribosomes[30]. We previously demonstrated that Asc1, the ortholog of the human *RACK1* translation inhibitor, promotes P-body formation during replication stress[8], further connecting P-bodies, DNA replication stress, and protein translation.

We also find a cohort of genes that when deleted suppress the HU sensitivity of P-body mutants, but that do not appear to be regulated by P-bodies at the level of transcript abundance. These mRNAs might represent targets of translational repression, rather than degradation, at P-bodies. Several lines of evidence connect regulation of translation to the DNA damage response. Translational repression of the tumor suppressor p53 can be mediated by the long noncoding RNA RoR during DNA damage[53] and by the Ku DNA-binding protein in unperturbed cells[54]. UV damage, ionizing radiation, and topoisomerase inhibition each result in a general inhibition of translation accompanied by selective translation of DNA damage response mRNAs[55–57]. Direct links between replication stress and protein translation are currently sparse, although levels of the MCM helicase are downregulated post-transcriptionally by replication stress[58]. It is clear, however, that extensive remodeling of protein abundance occurs during replication stress, and the majority of protein abundance changes are not explained by changes in mRNA abundance[8, 9]. Together, these observations raise the possibility that cellular translation capacity is regulated by P-body proteins during DNA replication stress.

Regulation of expression of the Yox1 transcriptional repressor is critical during the replication stress response. Overexpression of *YOX1* resulted in the downregulation of genes that are normally induced by HU treatment, suggesting that de-repression of Yox1 targets forms a large component of the transcriptional response to replication stress. Deletion of *YOX1* rescues the HU sensitivity of P-body mutants (*lsm1Δ*, *pat1Δ* and *xrn1Δ*). The simplest explanation for this rescue is that *YOX1* gene expression is excessive when P-bodies are dysfunctional, indicating that de-repression of *YOX1* targets is critical for cell fitness during replication stress. Surprisingly, we did not find many genes that are individually involved in HU resistance within the Yox1 transcriptional network. This observation led us to test whether the Yox1 regulon as a whole could exert an important role in HU resistance. Although we tested only a small subset of Yox1 targets, we readily detected double mutant combinations that cause HU sensitivity in the Yox1 transcriptional network, consistent with the idea that de-repression of more than one Yox1 target is required to maximize cell fitness during HU-stress. This possibility has been noted repeatedly in different DNA damage and replication stress-induced transcriptomes, where the overlap between individual regulated genes and genes that confer stress sensitivity when deleted individually is typically poor[4, 5, 59].

In addition to the role that the entire Yox1 regulon might exert to promote HU resistance, we also identified individual *YOX1* targets whose de-repression is critical to avoid replication stress-induced toxicity. In particular, *ALD6* encodes a key enzyme involved in acetaldehyde detoxification both under normal growth and during stress conditions[60]. Acetaldehyde can induce DNA inter-strand crosslinks that block DNA replication and transcription[46] and we found that acetaldehyde accumulation induces replication stress and greatly sensitizes cells to additional DNA replication stress induced HU. Interestingly, DNA inter-strand crosslink repair is essential for the survival of human cells experiencing endogenous replication stress[61] and aldehyde dehydrogenases like Ald6 are conserved throughout evolution, suggesting that accumulation of acetaldehyde could be an important contributor to DNA replication stress in all cells. Thus, the Yox1 regulon contributes to HU resistance both at the level of individual genes, and at the level of higher-order combinations of genes, consistent with transcriptional re-wiring in the absence of P-body proteins being an important element of the response to HU-induced DNA replication stress.

## Methods

**Yeast strains and growth conditions**. All strains used in this study are listed in Supplementary Table 4. *yox1Δ*, *yhp1Δ*, *xrn1Δ*, and *ald6Δ* strains were obtained from the yeast deletion collection[62]. Yeast were maintained on standard YPD or synthetic complete drop-out medium lacking the appropriate amino acids when necessary and grown at 30 °C unless specified otherwise. For overexpression experiments, yeasts were first grown to saturation in synthetic drop-out medium containing 2% raffinose as the only carbon and then diluted in fresh medium containing raffinose and 2% galactose, unless specified otherwise. All yeast strains were constructed in BY4741 or BY4742 backgrounds[63] using standard lithium acetate transformation. Strains expressing *YOX1* tagged with MS2 repeats were constructed as described[38].

**RNA extraction and sequencing for wild type and *lsm1Δ***. WT and *lsm1Δ* (JTY16) cells were grown to mid-logarithmic phase in YPD, sampled, and then treated with 200 mM HU, and sampled after 1, 2 and 4 h. Approximately 20 OD$_{600}$ of cell culture was used for RNA extraction. RNA extractions were performed using the yeast RiboPure kit (ThermoFisher Scientific) according to the manufacturer's instructions. Prior to sequencing library preparation, rRNAs were removed from the RNA samples using the yeast RiboZero kit (Illumina). Sequencing libraries were prepared using the TruSeq RNA Library Prep v2 kit (Illumina) and sequenced on a HiSeq2500 using v4 chemistry (Illumina). RNA-sequencing data were analyzed using the Tuxedo protocol[19]. Briefly, the 50 bp reads were aligned to the sacCer3 version of the *S. cerevisiae* genome with Tophat2[64] using the default parameters. Between 15 and 20 millions reads were obtained for every sample and alignment to the reference genome was achieved for more than 95% of the reads for every

sample. Genome annotations were downloaded from the SGD database (http://www.yeastgenome.org) on 7 October 2014. Transcripts were assembled with Cufflinks[19] and merged assembly files were generated with Cuffmerge[19]. Differential expression was assessed using Cuffdiff[19]. Default parameters were used to run Cufflinks, Cuffmerge, and Cuffdiff. Log2 ratios and statistical data generated with Cuffdiff were further analyzed using custom pipelines in R (https://www.r-project.org). Mean FPKM values, log2 ratios and $P$-values for every experiment are provided in Supplementary Data 1. Each RNA-sequencing experiment included two independent technical replicates. Correlation between replicates was always greater than 0.92. Raw sequencing data are accessible on GEO Omnibus with the accession number GSE83515. R scripts are provided in Supplementary Software 1.

To validate the results obtained using the Tuxedo analysis pipeline, RNA-Seq data were re-analyzed using EBSeq[23] or edgeR[24]. Raw sequencing count data generated by Cuffdiff before conversion to FPKM values were extracted for each gene and each RNA-Seq sample to create a matrix of raw counts compatible with EBSeq or edgeR analysis (Supplementary Data 1). EBSeq and edgeR analysis were performed using the respective R packages (EBSeq v1.16.0 and edgeR v3.18.1) downloaded from the Bioconductor website and according to the manual instructions.

**Protein and genetic interactions enrichment analysis.** Protein-protein and genetic interaction data were downloaded from BioGRID (http://thebiogrid.org/) on 24 August 2016. For each gene in the BioGRID data, a list of non-redundant interactors was generated. For each set of differentially expressed genes, the number of interactions involving all the genes in the given gene set (untreated or HU-core and increase or decrease in abundance) was determined and normalized to the number of genes in the given gene set. The same calculation was performed on the entire BioGRID dataset. $P$-values were calculated using the binomial test in R.

**GO term analysis.** GO term analysis was performed using the GO term finder tool (http://go.princeton.edu/) using a $P$-value cutoff of 0.01 and applying Bonferroni correction, querying biological process enrichment for each gene set. Transposon genes were removed from the analysis due to their repetitive nature. GO term enrichment results were further processed with REViGO[65] using the "Medium (0.7)" term similarity filter and simRel score as the semantic similarity measure. GO term network data generated by REViGO was loaded into Cytoscape v3.2.0[66] for visualization as shown in Fig. 1. Nodes were re-organized for clarity, and nodes representing terms with a frequency greater than 15% in the REViGO output were eliminated as too general. Gene lists used for the GO enrichment analyses, as well as the list of enriched GO terms obtained are provided in Supplementary Data 2.

**Synthetic genetic array-based suppressor screen.** *SGA procedure*: Strains RLKY6 (*lsm1Δ*) and RLKY15 (*pat1Δ*) (Supplementary Table 1) were crossed to the yeast deletion collection[62] and a collection of temperature sensitive mutants for a set of essential genes[27] in a condensed 1536 colony array format using a BioMatrix (S&P Robotics Inc.) automated pinning system. In this format, each mutant on the deletion and temperature sensitive arrays gave rise to one diploid colony for each query strain. After diploid selection, yeast diploid strains were sporulated and *MAT***a** haploid double-mutants were selected using standard SGA media[28]. After double mutant selection, the condensed 1536 array was expanded to a quad-ruplicated 1536 array (four colonies for each double mutant) and pinned on double mutant selection SGA medium once again. To identify suppressors of *lsm1Δ* and *pat1Δ* HU sensitivity, the quadruplicated 1536 arrays were first pinned on double mutant selection SGA medium. The newly pinned arrays were then immediately used as pinning source to re-pin the cells on double mutant selection SGA medium containing HU at concentrations of 100 and 150 mM for the deletion array and temperature sensitive array, respectively. Cells were grown for 3 days at 30 °C. Plates were photographed in order to determine the colony size computationally. The experiment was repeated three times, with the exception of *pat1Δ × xxx ts*, which was repeated twice.

*Calculation of fitness in HU*: We defined the fitness in HU as the ratio between the colony size in HU and the colony size in absence of drug. Raw colony size for each double mutant grown in absence or presence of HU was determined using SGATools[67] (http://sgatools.ccbr.utoronto.ca/) and further data analysis was perform in R using a customized pipeline. Double mutants displaying small colony size in the absence of drug were removed from the analysis. For each of the remaining double mutants, fitness in HU was calculated and normalized on a per plate basis using the $Z$-score method (difference between the colony size of a given mutant and the average colony size on the plate divided by the standard deviation observed for colony size on the plate). All mutants with an average $Z$-score, between three replicates, greater than or equal to +1 were considered as suppressor hits. All mutants with an average $Z$-score, between three replicates, less than or equal to −1 were considered as synthetic sick hits. Data are provided in Supplementary Data 3. The R script is provided in the Supplementary Software 1.

**Drug sensitivity assays.** Cells were grown to saturation in YPD or synthetic media containing raffinose and diluted in fresh medium at a cell concentration of ~$10^7$ cells/ml. Ten-fold serial dilutions (unless specified otherwise) of the fresh cell suspension were then spotted on the appropriate agar plates with or without drug and incubated at 30 °C unless specified otherwise. For acetaldehyde plate preparation, the drug was spread on 25 ml YPD plates immediately before spotting the cells. Plates were photographed after 3–4 days or 7 days of incubation for YPD and synthetic media plates respectively.

**Microscopy.** For *YOX1* mRNA imaging, cells were grown overnight in synthetic medium lacking methionine in order to induce expression of the MS2 coat protein and then diluted in low-fluorescence medium containing methionine 4–6 h prior to imaging. Where indicated, cells were treated with HU for 2 h. Cells were imaged on a Leica DMI6000 confocal microscope.

For Yox1-GFP protein imaging, cells were first grown in synthetic complete medium overnight in a 96-well plate and diluted the next day for a second overnight culture in low-fluorescence synthetic complete medium. Cells were finally diluted to an $OD_{600}$ of 0.05 in fresh low-fluorescence synthetic complete medium in a 384-well imaging plate 2 h prior to imaging and incubated at 30 °C. Cells were imaged on an Opera confocal fluorescence microscope (PerkinElmer) in absence of drug and 2 and 4 h after addition of HU at a final concentration of 200 mM.

For Rnr3-GFP protein imaging, cells in exponential phase grown in complete low-fluorescence medium for 17–19 h were imaged on a Leica DMI6000 confocal microscope.

**Image analysis.** For Yox1-GFP fluorescence quantification, image files from the Opera were analyzed using CellProfiler v2.1.1[68]. Cell objects or nucleus objects were segmented using the Rpl39-tdTomato and Hta2-mCherry signals, respectively. GFP, tdTomato, and mCherry intensities were then measured in the segmented object. Median intensity measurements generated with CellProfiler were then exported for further analysis in R using customized pipelines. Yox1-GFP median intensities in the segmented cells or nuclei were normalized to Rpl39-tdTomato or Hta2-mCherry median intensities, respectively. CellProfiler pipelines and the R script are included in the Supplementary Material.

For Rnr3-GFP fluorescence quantification, image files from the Volocity software v6.3 (PerkinElmer) were exported to tif format and analyzed using CellProfiler v2.1.1[68]. Cell objects were segmented using the Rpl39-tdTomato signal. GFP intensity was then measured in the segmented object. Median intensity measurements generated with CellProfiler were then exported for further analysis in R using customized pipelines. CellProfiler pipelines and the R script are included in the Supplementary Software 1.

**Protein extraction and immunoblot analysis.** Protein extraction and immuno-blotting was carried out as previously described[69]. Briefly, exponentially growing cells in YPD were collected before and after addition of HU at a final concentration of 200 mM. Cells were then fixed with trichloroacetic acid (10% final) for 15 min and whole-cells extracts prepared. Proteins were separated on a SDS–PAGE gel and immunoblots were performed using the FLAG-M2 antibody (Sigma, #F3165) diluted 1:5000 or PGK1 antibody (Novex, #4592250) diluted 1:1,000,000. An uncropped scan of the membrane is included in Supplementary Fig. 5.

**RNA extraction and sequencing for cells overexpressing *YOX1*.** Cells were grown to mid-logarithmic phase in synthetic complete medium lacking uracil and containing 2% raffinose, sampled, and then *YOX1* overexpression was induced by the addition of galactose to 2% final, cultures were treated with 200 mM HU, and sampled after 2 and 4 h. Approximately 20 $OD_{600}$ of cell culture was used for RNA extraction. RNA extractions, RNA-seq, and data analysis was performed as described above. Mean FPKM values, log2 ratios, and $P$-values are provided in Supplementary Data 4. Each RNA-sequencing experiment included two independent biological replicates. Correlation between replicates was always >0.92. Raw sequencing data are accessible on GEO Omnibus with the accession number GSE83515.

**qRT-PCR.** RNA extraction was performed as described for the RNA-Seq experiments. Reverse transcription was performed using the Superscript II Reverse Transcriptase (Thermo Fischer Scientific) on ~100 ng of total RNA with the following gene specific reverse primers at a final concentration of 0.25 µM: RLO62 (5′-ACGGGAGTCAACGTCTTCTT-3′) for *YOX1* mRNA, RLO198 (5′-TCTTGGC GCCTTCTTTCTTA-3′) for *ALD6* mRNA, and RLO25 (5′-CTTCTTAGCCT-TAGCCATTT-3′) for *RPL39* mRNA. qPCR was then performed on 2 µl of cDNA in a PCR reaction volume of 25 µl using the SYBR Green PCR Master Mix (Thermo Fischer Scientific) and with the following primer pairs at a final concentration of 0.1 µM: RLO61 (5′-CAAGGCGGACATACTTCGTG-3′) and RLO62 for *YOX1* mRNA, RLO197 (5′-ACGACGAACTATTGGCTGCT-3′) and RLO198 for *ALD6* mRNA, RLO24 (5′-GTGGTCCAGCGTGGTTTATG-3′) and RLO25 for *RPL39* mRNA. Differential expression was then assessed using the standard ΔCt method using *RPL39* mRNA as internal standard. *RPL39* standard was chosen as it was not differentially expressed in *lsm1Δ* cells or upon HU exposure in our RNA-Seq experiments.

**Growth kinetics**. Strains were first grown to saturation in liquid medium in a 96-well plate and then diluted to an $OD_{600}$ of 0.5 in fresh medium. An aliquot of 10 μl of this cell suspension was then used to inoculate 90 μl of fresh medium containing HU in a 96-well plate. Cell growth was monitored by measuring $OD_{600}$ every 15 min for 96 h using a Sunrise (Tecan) 96-well plate OD reader. $OD_{600}$ data were retrieved and further analyzed in R using a customized pipeline. The R pipeline is provided in the Supplementary Software 1. Briefly, cell growth was assessed by measuring the area under the growth curve ($OD_{600}$ as a function of time) and growth data were expressed as a percentage of growth for a given mutant compared to the control strain grown under the same conditions and during the same experiment. The experiment was repeated at least three times and included at least two independent biological replicates every time.

**HU sensitivity of Yox1 target double mutants**. *Double mutant construction and growth monitoring*: Deletion mutants, available in the yeast deletion collection[62] and corresponding to the identified Yox1 targets (*ald6Δ*, *ics2Δ*, *mmr1Δ*, *pry3Δ*, *rad54Δ*, *uth1Δ*, *yml131WΔ*) or the *his3Δ* control strain were crossed with the following SGA query strains: *ald6Δ* (RLKY105), *mmr1Δ* (RLKY108), *pry3Δ* (RLKY104), *rad54Δ* (RLKY106), *yml131WΔ* (RLKY107), and the *ura3Δ* control strain (RLKY109). After diploid selection and sporulation, haploid double mutants were selected using the SGA protocol[28]. *uth1Δ* and *ics2Δ* query mutants were excluded from the analysis as they did not display appropriate genetic linkage. Haploid double mutants were first grown to saturation in SGA double selection liquid medium and then diluted 1:100 in 90 μl of YPD or YPD containing HU at a concentration of 25 mM in a 96-well plate. Cell growth was then monitored by measuring $OD_{600}$ every 15 min for 24 h using a Genios (Tecan) 96-well plate OD reader. The experiment was performed 5 times.

*Genetic interaction assessment*: For each growth condition (YPD or YPD+HU) we calculated the fitness ($\varepsilon$) of each strain by measuring the area under the growth curve (OD600 as a function of time). The $\varepsilon$ value for each mutant was then normalized to the $\varepsilon$ value obtained for the control strain (*his3Δ ura3Δ*). Negative genetic interactions were then assessed by comparing the observed fitness for each double mutant to the expected fitness calculated by multiplying the observed fitness for each of the two individual single mutants. Statistical significance was assessed by comparing expected and observed fitness values using the Student's *t*-test. In order to assess whether genetic interactions were exacerbated by HU treatment, we tested whether the observed fitness (relative to WT) of each double mutant displayed a statistically supported lower fitness in HU than in the untreated condition. Fitness data and *P*-values are provided in Supplementary Data 5.

**Acetaldehyde–hydroxyurea drug interaction**. *Experimental design*: Drug interactions were tested as previously described, with minor modifications[70]. Each drug was prepared at a 2× concentration in YPD (acetaldehyde 1% and HU 200 mM). An aliquot of 50 μl of 2:3 serially diluted HU or 4:5 serially diluted acetaldehyde were then mixed together in 96-well plate in order to create a 6×6 array allowing testing of all drug combinations possible. In order to test the effect of each drug alone, serially diluted drugs were mixed with 50 μl of YPD. A no drug control was also included in the array design. An aliquot of 10 μl of a saturated yeast BY4741 culture, diluted to an $OD_{600}$ of 0.8, was added to each well, and growth was monitored by measuring $OD_{600}$ every 15 min for 24 h using a Genios (Tecan) 96-well plate OD reader. The experiment was performed three times.

*Data analysis*: For each condition (drug alone, drug combination, no drug), the fitness ($\varepsilon$) of yeast cells was evaluated by measuring the area under the growth curve (OD600 as a function of time). The $\varepsilon$ value for each condition was then normalized to the $\varepsilon$ value obtained for the yeast cells grown in absence of drug. The interaction between acetaldehyde and HU was then evaluated by substracting the expected fitness (calculated by multiplying the normalized fitness values obtained for each drug alone at a given concentration) by the observed fitness value in presence of the combination of the two drugs. In this setting, negative values indicate synergy between acetaldehyde and HU.

**Cell-cycle analysis**. Saturated cultures containing WT or *lsm1Δ* (JTY16) were diluted in fresh YPD or YPD containing HU at a final concentration of 200 mM. Samples for flow-cytometry were taken every hour during 4 h. When cell synchronization was required (*ald6Δ* experiment), G1 was induced for 3 h on exponentially growing cells using α-factor at a concentration of 2.5 μg/ml. G1 release was achieved by treating the culture with Pronase at a concentration of 100 μg/ml. Sample preparation for flow-cytometry analysis was performed as previously described[12]. Flow cytometric data were acquired on a FACSCanto II (BD Biosciences) and analyzed using the FlowJo software.

**Data availability**. All data generated or analyzed during this study are included in this published article (and its supplementary information files) except for the raw RNA-sequencing files that are available in the GEO Omnibus database with the accession number GSE83515 (https://www.ncbi.nlm.nih.gov/geo/). All the scripts generated to analyze the data during this study are included in the supplementary information files of this article.

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

## Acknowledgements

We thank Guihong Tan and Charlie Boone for generously providing the temperature sensitive yeast strain array. We thank Dax Torti and the Donnelly Sequencing Centre for RNA-Seq library preparation and sequencing. We are very grateful to all the members of the Brown lab for helpful discussions and to Craig Smibert for his critical reading of the manuscript. This work was supported by Canadian Institutes of Health Research grant MOP-137043 and by a Cancer Research Society grant (to G.W.B.).

## Author contributions

Conceptualization: R.L.-K. and G.W.B.; methodology: R.L.-K.; formal analysis: R.L.-K.; writing and original draft: R.L.-K.; writing, review, and editing: R.L.-K. and G.W.B.

## Additional information

**Competing interests:** The authors declare no competing financial interests.

