## [Peer Review File · Nature Communications]

Reviewers' Comments:

Reviewer #1 (Remarks to the Author)

This manuscript examines the functions of P-bodies in regulating cellular responses to replication stress. P-bodies are sites of mRNA decapping to decrease mRNA abundance. Stresses including replication stress induce P-body formation and function and P-body proteins are important for cell viability in response to these stresses. The authors utilized RNA sequencing and genetics to identify mRNAs that may be regulated in a P-body dependent manner to yield resistance to replication stress. Their analysis identified Yox1, a transcriptional repressor as one of several candidates and they validated Yox1 mRNA regulation by P-body processing as important for replication stress responses. Furthermore, they identified two gene targets of Yox1 regulation that contribute to these responses.

Overall, I found the data in the manuscript to be compelling and the conclusions interesting. While it is not surprising that Yox1 is involved in controlling gene expression in response to replication stress, the control of Yox1 by P-body dependent processing is novel and interesting. The only thing that I would have liked the authors to do was to demonstrate unequivocally that ALD6 and ICS2 are really direct targets of Yox1 important for stress responses. For example, it would be helpful to demonstrate direct binding of Yox1 protein to the putative Yox1 binding sites in the promoters of these genes. Also, monitoring the effects of deletion of these binding sites on gene expression and cell viability would be useful.

Reviewer #2 (Remarks to the Author)

This manuscript addresses the role of mRNA decay factors in the control of gene expression during DNA replication stress. The main point of the work is that Lsm1 and Pat1 play roles in shaping the transcriptome during HU stress, and that by keeping specific mRNAs expressed at lower levels, they contribute to successful DNA damage response. This conclusion is supported by, i) RNA-Seq of Wt and *lsm1Δ* strains identifying mRNAs that are overexpressed either normally or in HU, ii) showing that deletion of 6 of these overexpressed mRNAs can partially suppress the HU sensitivity of *lsm1Δ* or *pat1Δ* strains, iii) focusing on YOX1, a transcriptional repressor, as key regulator whose over-expression in *pat1Δ* and *lsm1Δ* strains leads to HU sensitivity. They then go on and show the Yox1 mRNA can be localized to P-bodies, and that the Yox1 protein goes up a little bit in *lsm1Δ* with or without HU and shows increased nuclear localization. Finally, they show that overexpression of Yox1 via galactose induction, or deletion of the Lsm1 gene leads to changes in the expression of genes that can contribute to the sensitivity of the DNA damage response. However, taken together these observations are not striking because, a) broadly, a role for mRNA decay in regulating gene expression is well established, and b) an actual mechanistic role for any of the downstream targets in mediating HU-related stress (which might not be DNA replication stress- please see comment 5 below) is lacking in the manuscript. Hence, the insights on mRNA decay/P-bodies playing a role in gene expression during replicative stress not sufficient to warrant publication in Nature Communications, unless the specific targets and regulatory loops identified in this manuscript are important to the DNA-damage community.

Specific comments:

1. Expression of several genes increased as well as decreased in the RNA-seq dataset as a function of HU stress in Wt vs *lsm1Δ*. One confounding issue with such analyses is that if some mRNA increase in abundance, they take up more sequence space, as a result some mRNAs get underrepresented and register as reduced levels. Is the subset of decreased mRNA in these datasets controlled for this eventuality?
2. An additional limitation of the RNA-seq data is the lack of target validation with an alternative technique such as northern blotting or RT-PCR. Such an additional validation can strengthen the argument for changes in expression of the identified target genes.

3. On that note, the authors model a role for P-bodies in regulating the level of Yox1 mRNA, yet the actual levels of total Yox1 mRNA in *lsm1Δ* and *pat1Δ* have not been measured.
4. Are these changes truly because of mRNA decay? Steady-state levels in gene expression don't often change much even with a change in decay rates. Do deletions in other mRNA decay genes yield the same changes in levels of targets identified?
5. Are the effects observed in gene expression, genetic suppression etc., due to HU-related stress or DNA replication stress, specifically? Is the expression of the identified targets affected in a similar manner upon exposure to other agents that lead to DNA replication stress?
6. One concern is whether the effects in gene expression observed in *lsm1Δ* (or *pat1Δ*) are due to P-bodies. It is my understanding that *lsm1Δ* and *pat1Δ* strains do not prevent P-body assembly. While this does not impact the key observations that these proteins can affect gene expression during DNA stress, it does affect the suggested role for P-bodies per se.
7. Figure 4.- nRNA should be mRNA on Y-axis.
- 8) Why is *td-tomato* used as control in Figure 5a? Shouldn't the controls be done in +/- HU to show how that alteration affects the signal?

Reviewer #3 (Remarks to the Author)

The manuscript characterizes mRNAs associated with p-bodies caused by HU-induced replication stress. The transcriptional repressor Yox1 is identified as localizing in P-bodies, and accumulates in the nucleus of *lsm1*-mutant cells. In general, the manuscript makes great strides in characterizing the function of P-bodies. I think the manuscript is beautifully written and the figures look very clean and professional.

The subject matter seems to appeal to a specific readership because it combines DNA-replication, P-bodies, and hydroxyurea stress. Nevertheless, it is written in a very accessible way, and could appeal to a broad readership given that it may reveal some fundamental biology of P-bodies.

In summary, I would accept this manuscript. I point out a few places where the manuscript could be improved.

A few minor formatting issues such as two periods after the sentence:

...exonuclease Xrn1, which together determine the decapping or degradation rate of mRNAs..

There is a super-script error in the sentence "Both RAD54 and RAD51 are induced during the DNA replication stress response or upon X-ray exposure 8; and our data,54. ".

Minor issues:

This particular sentence:

"Amazingly, we could identify specific YOX1 targets whose de-repression is critical to avoid replication stress induced toxicity, ALD6 and ICS2."

has two issues. First, "Amazingly" is a very strong word, and might be too strong for a publication. Would "Surprisingly" be more appropriate? Secondly, "replication stress induced" functions as an adjective, and should be hyphenated as "replication stress-induced toxicity".

Suggestions to the authors for improvement of readability:

How are "fitness values" defined? In the methods I see that it is the ratio of colony size in HU vs no drug. Would it be more clear to briefly define fitness when first introduced in the Results on page 6, or at least mention that it is defined by colony size?

The section title "Suppressors of replication stress sensitivity of P-body mutants" seems grammatically strange to me.

Concerning the statement: "Alternatively, absence of LSM1 could stabilize transcriptional repressors, resulting indirectly in mRNA abundance decreases, as has been observed in cells lacking the 5'-3' RNA exonuclease Xrn1"

This is an interesting hypothesis. I think it would strengthen this point to identify specific transcriptional repressors in the set of transcripts that increase in lsm1-mutants (other than Yox1) and mention here, although more detail is given in the discussion on this point. Along those lines, the statement in the manuscript about these differentially expressed genes does not state that the list of the 333 up- and 258 down-regulated genes in lsm1-mutants (in the absence of HU-stress) is also part of Table S2. As far as I can tell, Table S2 is only introduced in the ms in the context of HU-treatment. It would be helpful to reference this table when discussing these genes at the end of page 3.

Point-by-point responses:

Reviewer #1:

This manuscript examines the functions of P-bodies in regulating cellular responses to replication stress. P-bodies are sites of mRNA decapping to decrease mRNA abundance. Stresses including replication stress induce P-body formation and function and P-body proteins are important for cell viability in response to these stresses. The authors utilized RNA sequencing and genetics to identify mRNAs that may be regulated in a P-body dependent manner to yield resistance to replication stress. Their analysis identified Yox1, a transcriptional repressor as one of several candidates and they validated Yox1 mRNA regulation by P-body processing as important for replication stress responses. Furthermore, they identified two gene targets of Yox1 regulation that contribute to these responses.

Overall, I found the data in the manuscript to be compelling and the conclusions interesting. While it is not surprising that Yox1 is involved in controlling gene expression in response to replication stress, the control of Yox1 by P-body dependent processing is novel and interesting. The only thing that I would have liked the authors to do was to demonstrate unequivocally that ALD6 and ICS2 are really direct targets of Yox1 important for stress responses. For example, it would be helpful to demonstrate direct binding of Yox1 protein to the putative Yox1 binding sites in the promoters of these genes. Also, monitoring the effects of deletion of these binding sites on gene expression and cell viability would be useful.

We agree that demonstrating a direct binding of Yox1 on *ALD6* or *ICS2* promoters would be of interest. However, it is a reasonable possibility that Yox1 could regulate *ALD6* or *ICS2* expression indirectly (for example by repressing *ALD6* or *ICS2* transcriptional activator) without changing the conclusions of our study. Yox1 or Mcm1 binding sites are present in multiple copies in both promoters, but Yox1 binding has not been detected in high throughput studies. Since direct binding by Yox1 is not an essential component of our model, we have modified the manuscript to clarify that Yox1 regulation could be indirect: *We identified binding sites for both Yox1 and its co-repressor Mcm1 in the 1000-bp promoter regions of ALD6 and ICS2 using YeTFaSCo⁴⁴ (Table S9), although it is also possible that both are indirect targets. (p.11)*

Reviewer #2

Specific comments:

1. Expression of several genes increased as well as decreased in the RNA-seq dataset as a function of HU stress in Wt vs *lsm1Δ*. One confounding issue with such analyses is that if some mRNA increase in abundance, they take up more sequence space, as a result some mRNAs get underrepresented and register as reduced levels. Is the subset of decreased mRNA in these datasets controlled for this eventuality?

We agree that upregulation of hundreds of genes could take up more sequencing space and artificially decrease abundance of other transcripts. While this is a known limitation of normalization by RPKM, most current methods to identify differentially expressed genes from RNA-seq data (including Cuffdiff, which is what we used) apply more sophisticated normalization routines to overcome the limitation. It is true that different analysis methods can produce some differences in results, so we re-analyzed our RNA-seq data using two alternative

RNA-Seq data analysis methods: EBSeq and edgeR. In particular, EBSeq uses a Bayesian statistics approach, which takes into account the compositional structure of RNA-seq data. edgeR uses an alternative normalization method as compared to cuffdiff (our initial analysis method). Using both methods, we were able to confirm our conclusions. The comparison of the three analysis methods has been added (Supplemental Table S3), and the text has been modified on p.3: *Finally, to confirm that the differentially expressed genes that we identified were independent of the data analysis method used, we applied two different analyses of the RNA-Seq data to identify differentially expressed genes: EBSeq²³ and edgeR²⁴. Between 34 and 79% of the genes identified in our initial analysis were also identified using EBSeq or edgeR, depending on the time point analyzed (Table S3).*

And on p.6: *Independent reconstruction of the *pat1Δ* and *lsm1Δ* double mutants with each of the 11 genes resulted in validation of 6 putative target genes: *ARL3*, *ACF4*, *HHT1*, *TMA1*, *RRS1* and *YOX1*. Increased mRNA abundance in HU for 5 of these transcripts was confirmed by two independent data analysis methods. *ACF4* was confirmed by edgeR but not by EBSeq (Table S6).*

We also note that the correlation between biological replicates in our RNA-Seq experiments was very high ($R > 0.92$ for biological replicates, mentioned in the Methods section).

2. An additional limitation of the RNA-seq data is the lack of target validation with an alternative technique such as northern blotting or RT-PCR. Such an additional validation can strengthen the argument for changes in expression of the identified target genes.

We addressed the reviewer's concern by validating *YOX1* up-regulation and *ALD6* down-regulation in *lsm1Δ* cells using qRT-PCR. The data are presented in Supplemental Figures S4 and S5.

3. On that note, the authors model a role for P-bodies in regulating the level of Yox1 mRNA, yet the actual levels of total Yox1 mRNA in *lsm1Δ* and *pat1Δ* have not been measured.

We have now measured *YOX1* mRNA in both *lsm1Δ* and *pat1Δ* by qRT-PCR (Figures S4 and S5), in addition to the original measurement in *lsm1Δ* by RNA-seq.

4. Are these changes truly because of mRNA decay? Steady-state levels in gene expression don't often change much even with a change in decay rates. Do deletions in other mRNA decay genes yield the same changes in levels of targets identified?

We found that *YOX1* mRNA increases in *pat1Δ* and *xrn1Δ* cells (Figure S4 and text on p. 9 (*xrn1Δ*:wildtype = 3.7 ± 1.8)). Pat1 is an mRNA decapping protein, and Xrn1 is the predominant 5' to 3' exoribonuclease, indicating a role for mRNA decay functions in the reduction in *YOX1* mRNA abundance.

5. Are the effects observed in gene expression, genetic suppression etc., due to HU-related stress or DNA replication stress, specifically? Is the expression of the identified targets affected in a similar manner upon exposure to other agents that lead to DNA replication stress?

We tested whether the *lsm1Δ* differentially expressed genes in HU overlapped with genes whose expression is affected during DNA replication induced by treatment with MMS and found good

overlap (as much as 53%, depending on the dataset) suggesting that the transcriptional program that we identified is likely a response to DNA replication stress in general and not only HU-specific. The text has been modified on p.4: *The correlation with data obtained using a distinct replication stress agent, MMS, indicates that a substantial fraction of the transcriptional program that we identified is due to DNA replication stress (Fig. S2b,c).*

6. One concern is whether the effects in gene expression observed in *lsm1Δ* (or *pat1Δ*) are due to P-bodies. It is my understanding that *lsm1Δ* and *pat1Δ* strains do not prevent P-body assembly. While this does not impact the key observations that these proteins can affect gene expression during DNA stress, it does affect the suggested role for P-bodies per se.

Deletion of *LSM1* induces Dcp1, Dcp2, Edc3, Xrn1 and Dhh1 foci formation (due to the accumulation of RNA in the cytoplasm) and reduces Pat1 foci formation (see Teixeira & Parker, 2007, Mol Biol Cell). Deletion of *PAT1* prevents the formation of P-body granules for almost all core P-body proteins, including Lsm1 (see Teixeira & Parker, 2007, Mol Biol Cell). Given that *YOX1* mRNA abundance increases in *lsm1Δ* and *pat1Δ* cells, we suggest that the regulation of P-body formation is required for the regulation of *YOX1* mRNA abundance.

7. Figure 4.- nRNA should be mRNA on Y-axis.

This has been corrected.

8. Why is td-tomato used as control in Figure 5a? Shouldn't the controls be done in +/- HU to show how that alteration affects the signal?

Td-tomato was not used as a control in Fig. 5a. We used Hta2-mCherry as a nuclear marker to segment the nuclei in order to quantify nuclear and cytoplasmic Yox1-GFP. Both RFP and GFP channels are shown in both conditions (-/+ HU) in Fig. 5a.

Reviewer #3 (Remarks to the Author):

The manuscript characterizes mRNAs associated with p-bodies caused by HU-induced replication stress. The transcriptional repressor Yox1 is identified as localizing in P-bodies, and accumulates in the nucleus of *lsm1*-mutant cells. In general, the manuscript makes great strides in characterizing the function of P-bodies. I think the manuscript is beautifully written and the figures look very clean and professional.

The subject matter seems to appeal to a specific readership because it combines DNA-replication, P-bodies, and hydroxyurea stress. Nevertheless, it is written in a very accessible way, and could appeal to a broad readership given that it may reveal some fundamental biology of P-bodies.

In summary, I would accept this manuscript. I point out a few places where the manuscript could be improved.

A few minor formatting issues such as two periods after the sentence:

...exonuclease Xrn1, which together determine the decapping or degradation rate of mRNAs..
This has been corrected.

There is a super-script error in the sentence “Both RAD54 and RAD51 are induced during the DNA replication stress response or upon X-ray exposure 8; and our data,54. “.
This has been corrected.

Minor issues:

This particular sentence:

“Amazingly, we could identify specific YOX1 targets whose de-repression is critical to avoid replication stress induced toxicity, ALD6 and ICS2.“ has two issues. First, “Amazingly” is a very strong word, and might be too strong for a publication. Would “Surprisingly” be more appropriate? Secondly, “replication stress induced” functions as an adjective, and should be hyphenated as “replication stress-induced toxicity”.
The text has been corrected as suggested

Suggestions to the authors for improvement of readability:

How are “fitness values” defined? In the methods I see that it is the ratio of colony size in HU vs no drug. Would it be more clear to briefly define fitness when first introduced in the Results on page 6, or at least mention that it is defined by colony size?

This has been clarified in the main text. The manuscript now reads:

We then assessed the fitness of every double mutant, by measuring and comparing colony size in the presence and absence of HU, in triplicate.

The section title “Suppressors of replication stress sensitivity of P-body mutants” seems grammatically strange to me.

The title has been changed as follows

Suppressors of the replication stress sensitivity of P-body mutants

Concerning the statement: “Alternatively, absence of LSM1 could stabilize transcriptional repressors, resulting indirectly in mRNA abundance decreases, as has been observed in cells lacking the 5’-3’ RNA exonuclease Xrn1.” This is an interesting hypothesis. I think it would strengthen this point to identify specific transcriptional repressors in the set of transcripts that increase in lsm1-mutants (other than Yox1) and mention here, although more detail is given in the discussion on this point.

We looked whether there were up-regulated repressors and whether their targets were enriched in the subset of *lsm1*Δ down-regulated genes at the same time points but did not find repressors that showed this pattern, with the exception of *YOXI*.

Along those lines, the statement in the manuscript about these differentially expressed genes does not state that the list of the 333 up- and 258 down-regulated genes in *lsm1*-mutants (in the absence of HU-stress) is also part of Table S2. As far as I can tell, Table S2 is only introduced in the ms in the context of HU-treatment. It would be helpful to reference this table when discussing these genes at the end of page 3.

We added an earlier reference to Table S2 as suggested.

Reviewers' Comments:

Reviewer #1:

Remarks to the Author:

If the authors were able to show that it was direct and map the responsible DNA element, those mechanistic insights would significantly strengthen the overall conclusions. However, I agree with the author's response that the model does not require direct binding and the results would still be of interest.

Reviewer #2:

Remarks to the Author:

The authors have presented several lines of evidence to solidify claims made in the initial version of this manuscript. Specifically, the additional data presented are sufficient to address technical concerns raised previously.

There is one major issue that I still find problematic. Specifically, I disagree with the claim made by the authors regarding the role for "P-bodies" per se in regulating levels of transcripts, such as Yox1 in vivo. Measurable, yet insufficient effects of single gene deletions on abrogation of P-body assembly is well documented since the manuscript by Teixeira and Parker 2007 (e.g., Buchan et al., 2008, JCB). Furthermore, the change in mRNA levels and P-body assembly are correlative, and not causative. As result, a model for P-bodies in DNA replication stress response by "rewiring" transcriptome is not supported by the data, and most likely is incorrect. Certainly, the proteins found in P-bodies can have an effect, but whether it is P-body assembly per se has not been demonstrated.

I recommend one of two things: A) I suggest that the authors change the title and the tone of the manuscript such that the title and tone does not overstate the observations, which implicates P-bodies in regulating the mRNA, or B) Examine how *edc3Δ* or *edc3Δ lsm4Δc* strains, which have very strong effects on P-bodies (Decker et al., 2007, JCB), affect this process. If they also have a strong effect, then I would be more convinced P-bodies per se are involved in the response.

Reviewer #2 Remarks to the Author

The authors have presented several lines of evidence to solidify claims made in the initial version of this manuscript. Specifically, the additional data presented are sufficient to address technical concerns raised previously.

There is one major issue that I still find problematic. Specifically, I disagree with the claim made by the authors regarding the role for “P-bodies” per se in regulating levels of transcripts, such as Yox1 in vivo. Measurable, yet insufficient effects of single gene deletions on abrogation of P-body assembly is well documented since the manuscript by Teixeira and Parker 2007 (e.g., Buchan et al., 2008, JCB). Furthermore, the change in mRNA levels and P-body assembly are correlative, and not causative. As result, a model for P-bodies in DNA replication stress response by “rewiring” transcriptome is not supported by the data, and most likely is incorrect. Certainly, the proteins found in P-bodies can have an effect, but whether it is P-body assembly per se has not been demonstrated.

I recommend one of two things: A) I suggest that the authors change the title and the tone of the manuscript such that the title and tone does not overstate the observations, which implicates P-bodies in regulating the mRNA, or B) Examine how *edc3Δ* or *edc3Δ lsm4Δc* strains, which have very strong effects on P-bodies (Decker et al., 2007, JCB), affect this process. If they also have a strong effect, then I would be more convinced P-bodies per se are involved in the response.

As the reviewer seems to be satisfied with most of our previous response and revision, we have focused on the following: “There is one major issue that I still find problematic. Specifically, I disagree with the claim made by the authors regarding the role for “P-bodies” per se in regulating levels of transcripts, such as Yox1 in vivo.”

As the reviewer noted in his/her first review, whether the effects correlate exactly with formation of P-body granules does not impact the key observations that the P-body components we tested, *Lsm1*, *Pat1*, and *Xrn1*, affect gene expression during DNA replication stress.

Nonetheless, we directly addressed the concern that the effect on gene expression that we observe might not be due to P-body granules per se in our response to the original review, by examining the effect of *pat1*Δ, which has a very strong effect on P-body formation (as shown by the Parker lab in glucose deprivation [PMID: 17429074]). The Buchan et al paper describes the same effect of *pat1*Δ described in the earlier Parker paper. More importantly, our 2012 study [PMID: 22842922] found that P-bodies do not form in *pat1*Δ during replication stress, which is of course the relevant condition for our present study. We understand the reviewer's point that some subset of P-body proteins still can form granules in the absence of Pat1, but these are not functional P-bodies as they lack de-capping activity [PMID: 17429074]. The reviewer focused on the *edc3*Δ or *edc3*Δ*lsm4*Δc mutants described in Buchan et al, perhaps because they give a more complete reduction of P-bodies during glucose deprivation than does *pat1*Δ (although the effect of *pat1*Δ on P-body formation is abundantly clear when either Lsm1-GFP or Edc3-GFP are examined (Teixeira and Parker, 2007)). However, in the relevant HU replication stress condition, *pat1*Δ eliminates P-bodies (as does *edc3*Δ). Both *PAT1* and *EDC3* promote decapping, and so interrogate the same pathway. Finally, the *edc3*Δ*lsm4*Δc mutant combo tends to destabilize mRNAs (PMID: 27543059). While we also present evidence that in many cases mRNAs can also be destabilized in P-body mutants, the transcript that we focus on, *YOX1*, belongs to the class of mRNAs that are stabilized in P-body mutants. To summarize, we have correlated P-body defects with *YOX1* mRNA abundance in several ways, and the experiment that the reviewer proposes is at best equivalent to one of the experiments (*pat1*Δ) that we have already performed. *To address the reviewer's concern, we have more explicitly highlighted the implications of the pat1Δ analysis with respect to the correlations between P-bodies and YOX1 expression, in the text.*

The question of whether the P-body granules themselves are required for the functions of P-body proteins in mRNA decay is under active debate in the P-body field, and there are examples both where granule formation correlates and where it does not. It is also the case that not all cytoplasmic granules containing P-body proteins are active P-bodies. For example, some of the P-body components form granules in the absence of Lsm1, but these are not functional P-bodies as they lack de-capping activity [PMID: 17429074]. We provide three separate pieces of evidence that P-bodies per se are important in our case: 1. The increase in *YOX1* mRNA occurs in *lsm1*Δ, where P-bodies are not functional; 2. The increase in *YOX1* mRNA occurs in *pat1*Δ, where P-bodies do not form during replication stress; and 3. The *YOX1* mRNA localizes to cytoplasmic granules that contain P-body proteins and accumulate, as do P-bodies, when *xrn1* is deleted, and which we therefore conclude are P-bodies. The simplest interpretation of our data is that *YOX1* is degraded at P-bodies during replication stress. Our interpretation does not exclude other, more complicated models, in which the three functionally distinct P-body components that we examine, Lsm1, Pat1, and Xrn1, also degrade *YOX1* mRNA at sites external to P-bodies. *We have added a discussion of the possibility that the effect of P-body components on YOX1 expression could take place at locales external to visible P-body granules. In line with 'option A' proposed by reviewer 2, we have clarified that our data implicate P-bodies in regulating YOX1 mRNA but that assembly of visible P-body granules per se might not be absolutely required.*

I think we have made a thorough and good-faith effort to address all of Reviewer 2's concerns in detail, so I hope you will agree that the manuscript is ready for publication.

Sincerely,
Grant W. Brown, Ph.D.